# Temporally specific gene expression and chromatin remodeling programs regulate a conserved *Pdyn* enhancer

**Robert A Phillips[1], Ethan Wan[1], Jennifer J Tuscher[1], David Reid[1], Olivia R Drake[1], Lara Ianov[1,2], Jeremy J Day[1]\***

[1]Department of Neurobiology, University of Alabama at Birmingham, Birmingham, United States; [2]Civitan International Research Center, University of Alabama at Birmingham, Birmingham, United States

**Abstract** Neuronal and behavioral adaptations to novel stimuli are regulated by temporally dynamic waves of transcriptional activity, which shape neuronal function and guide enduring plasticity. Neuronal activation promotes expression of an immediate early gene (IEG) program comprised primarily of activity-dependent transcription factors, which are thought to regulate a second set of late response genes (LRGs). However, while the mechanisms governing IEG activation have been well studied, the molecular interplay between IEGs and LRGs remain poorly characterized. Here, we used transcriptomic and chromatin accessibility profiling to define activity-driven responses in rat striatal neurons. As expected, neuronal depolarization generated robust changes in gene expression, with early changes (1 hr) enriched for inducible transcription factors and later changes (4 hr) enriched for neuropeptides, synaptic proteins, and ion channels. Remarkably, while depolarization did not induce chromatin remodeling after 1 hr, we found broad increases in chromatin accessibility at thousands of sites in the genome at 4 hr after neuronal stimulation. These putative regulatory elements were found almost exclusively at noncoding regions of the genome, and harbored consensus motifs for numerous activity-dependent transcription factors such as AP-1. Furthermore, blocking protein synthesis prevented activity-dependent chromatin remodeling, suggesting that IEG proteins are required for this process. Targeted analysis of LRG loci identified a putative enhancer upstream of *Pdyn* (prodynorphin), a gene encoding an opioid neuropeptide implicated in motivated behavior and neuropsychiatric disease states. CRISPR-based functional assays demonstrated that this enhancer is both necessary and sufficient for *Pdyn* transcription. This regulatory element is also conserved at the human *PDYN* locus, where its activation is sufficient to drive *PDYN* transcription in human cells. These results suggest that IEGs participate in chromatin remodeling at enhancers and identify a conserved enhancer that may act as a therapeutic target for brain disorders involving dysregulation of *Pdyn*.

## eLife assessment

This is an **important** study that uses chromatin accessibility as a measure to determine the impact of neuronal activity on the state of chromatin regulatory elements in striatal neurons. The authors provide **convincing** evidence of how Pdyn gene expression is highly dependent on a distal regulatory genomic region both at basal and upon neuronal activation in this particular system, a mechanism conserved as well in human neuronal cells. Although the basic idea of accessibility changes have been studied before, this paper ties previous findings all together in one place and uses the analysis to identify a functionally relevant and conserved enhancer for the prodynorphin gene with potential relevance for neuropsychiatric disorders beyond basic cellular neuroscience. The study will

**\*For correspondence:**
jjday@uab.edu

be of interest to neuroscientists studying the striatum, neuronal plasticity, or related neuropsychiatric disorders.

## Introduction

Experience-dependent cellular adaptations within the brain circuits that control motivated behaviors are critical for learning, memory, and long-term behavioral change. In psychiatric disorders such as drug addiction, these cellular changes are hijacked to drive maladaptive behavioral changes that promote drug seeking (*Mews et al., 2018*; *Adinoff, 2004*). Furthermore, mutations that alter the function of activity-dependent transcription factors have been implicated in a host of neurodevelopmental and autism spectrum disorders (*Ebert and Greenberg, 2013*; *Zhang et al., 2016*). Adaptations to novel stimuli are regulated by temporally and functionally distinct activity-dependent transcriptional programs. For example, various forms of neuronal activation result in the rapid induction of immediate early genes (IEGs), including transcription factors such as *Fos* (aka c-Fos), *Npas4*, and *Nr4a2*. These genes follow a temporally dynamic profile, with elevated expression within 1 hr of a stimulus and rapid return to baseline levels. In contrast, the same stimuli promote expression of a more delayed gene expression program, termed late response genes (LRGs) (*Yap and Greenberg, 2018*; *Tyssowski et al., 2018*). This set of genes includes kinases, neurotrophic factors, and neurotransmitter receptors. Current models of activity-dependent gene expression suggest that these distinct transcriptional waves work together to promote enduring cellular and behavioral adaptations.

While genes within the IEG expression program are required for cellular and behavioral changes following stimulation, the exact mechanisms by which they contribute to LRG expression and the functional consequences of this process remain poorly characterized. Recent evidence suggests that chromatin remodeling at genomic enhancers is a key event linking LRGs to IEGs (*Vierbuchen et al., 2017*; *Su et al., 2017*). In non-neuronal systems, AP-1, an activity-dependent transcription factor consisting of *Fos* and *Jun* family members, directly contributes to chromatin remodeling at LRG enhancers (*Vierbuchen et al., 2017*; *Hrvatin et al., 2018*). However, despite ample evidence for activity-dependent transcription of AP-1 members in multiple brain regions and cell types, understanding the nature and functional consequences of LRG induction remains a challenge for several reasons. First, emerging evidence has revealed that different classes of neurons induce distinct LRG programs in response to the same neuronal activation (*Hrvatin et al., 2018*; *Spiegel et al., 2014*; *Hu et al., 2017*; *Roethler et al., 2023*; *Gray et al., 2015*; *Gallegos et al., 2022*). Second, different types of stimuli may give rise to non-overlapping constellations of IEG transcription factors, which could promote activation of distinct LRGs to tune neuronal responses (*Tyssowski et al., 2018*; *Lin et al., 2008*). Finally, even where chromatin remodeling has been identified at candidate enhancers near LRGs (*Vierbuchen et al., 2017*; *Malik et al., 2014*), the consequences of this remodeling have not been concretely linked to transcriptional activation of candidate LRGs.

Here, we used next-generation sequencing approaches to characterize temporally distinct, activity-dependent transcriptomic and epigenomic reorganization in cultured rat embryonic striatal neurons, an in vitro model containing many cell types implicated in learning, motivation, reward, and substance use disorders (*Savell et al., 2020*). These experiments comprehensively characterized activity-responsive genes in both the IEG and LRG expression programs in striatal neurons, and revealed a temporal decoupling between IEG activation and activity-dependent chromatin remodeling. Further functional studies suggest that translation of IEGs is necessary for activity-dependent chromatin remodeling, and that sites of chromatin opening are enriched for AP-1 transcription factor motifs. Combining transcriptional and epigenomic profiling allowed us to identify a putative enhancer upstream of the prodynorphin (*Pdyn*) gene locus that is conserved in the human genome. CRISPR-based activation and repression experiments provided functional validation that this region serves as a *Pdyn* enhancer in both rat neurons and dividing human cell lines. Collectively, these results highlight the mechanisms through which neuronal activity promotes gene expression changes to modify neuronal function, and have relevance for brain disease states characterized by alterations to this process.

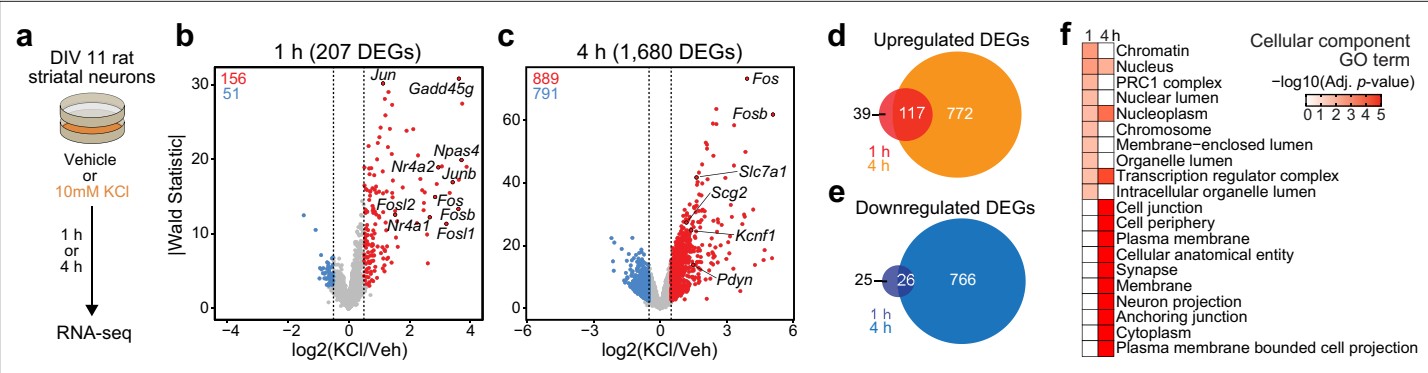

**Figure 1.** Characterization of temporally and functionally distinct activity-dependent gene expression programs in cultured striatal neurons. (**a**) Experimental design. DIV11 cultures were depolarized for 1 or 4 hr with 10 mM KCl. Following treatment, RNA-seq libraries were constructed. (**b, c**) Volcano plots displaying gene expression changes after 1 and 4 hr of neuronal depolarization. (**d, e**) Venn diagrams comparing 1 and 4 hr up- and downregulated differentially expressed genes (DEGs). (**f**) Top 10 cellular component GO terms for 1 and 4 upregulated DEGs.

The online version of this article includes the following source data and figure supplement(s) for figure 1:

**Figure supplement 1.** Striatal neuron immediate early genes (IEGs) and late response genes (LRGs) are temporally and functionally distinct and are induced by a variety of stimuli.

**Figure supplement 1—source data 1.** RT-qPCR data.

**Figure supplement 1—source data 2.** RT-qPCR data.

# Results

## Characterization of temporally and functionally distinct transcriptional programs following neuronal depolarization

Activity-dependent transcriptomic and epigenomic reorganization has been heavily implicated in neuropsychiatric diseases, such as drug addiction. Previously, our laboratory established cultured rat embryonic striatal neurons as an in vitro model for studying activity-dependent processes as these cultures contain the same cell types affected by drugs of abuse in the rat ventral striatum (*Savell et al., 2020*). To characterize IEG and LRG expression programs in this model system, we performed RNA-seq following neuronal depolarization with 10 mM KCl for 1 or 4 hr (*Figure 1a*). Following 1 hr of depolarization, we identified 207 differentially expressed genes (DEGs; defined as genes with an adjusted p-value <0.05 and |log2FoldChange| > 0.5). Notably, ~75% of these genes were upregulated by KCl, including activity-dependent transcription factors such as *Npas4*, *Fos*, *Fosb*, *Fosl2*, and *Nr4a1* (*Figure 1b*, *Supplementary file 1*). In contrast, 4 hr of depolarization resulted in significant transcriptomic reorganization, with 1680 genes identified as DEGs (*Figure 1c*, *Supplementary file 1*). DEGs upregulated by KCl at this timepoint included the opioid propeptide *Pdyn*, as well as the voltage-gated potassium channel *Kcnf1* (*Figure 1c*). Interestingly, overlap between 1 hr DEGs (putative IEGs) and 4 hr DEGs (putative LRGs) primarily occurred when IEGs such as *Fos* were upregulated at both 1 and 4 hr (*Figure 1d and e*; *Figure 1—figure supplement 1*). In contrast, most 4 hr DEGs such as *Pdyn* demonstrated temporally specific upregulation and were only activated following 4 hr of depolarization (*Figure 1—figure supplement 1b*).

To determine whether 1 and 4 hr specific DEGs were functionally distinct, we used gProfiler (*Raudvere et al., 2019*) to identify enriched cellular component and molecular function gene ontology (GO) terms in each gene list. 1 hr DEGs, or IEGs, were enriched for cellular component terms such as 'Chromatin', 'Chromosome', 'Nucleus', and 'Transcription regulator complex' (*Figure 1f*), suggesting that most DEGs were transcription factors or genes encoding proteins involved in transcriptional regulation. While not in the top 10 cellular component GO terms, 'Nucleus', 'Nucleoplasm', and 'Transcription regulatory complex' are also significantly enriched in the 4 hr DEGs. However, 4 hr DEGs, or LRGs, were also enriched for cellular component terms such as 'Synapse' and 'Neuron projection' (*Figure 1f*), suggesting that these DEGs encode proteins required for cellular adaptations to stimulation. Molecular function GO term analysis found overlap between IEGs and LRGs (*Figure 1—figure supplement 1c*), with both IEG-specific and overlapping molecular function GO terms including transcription

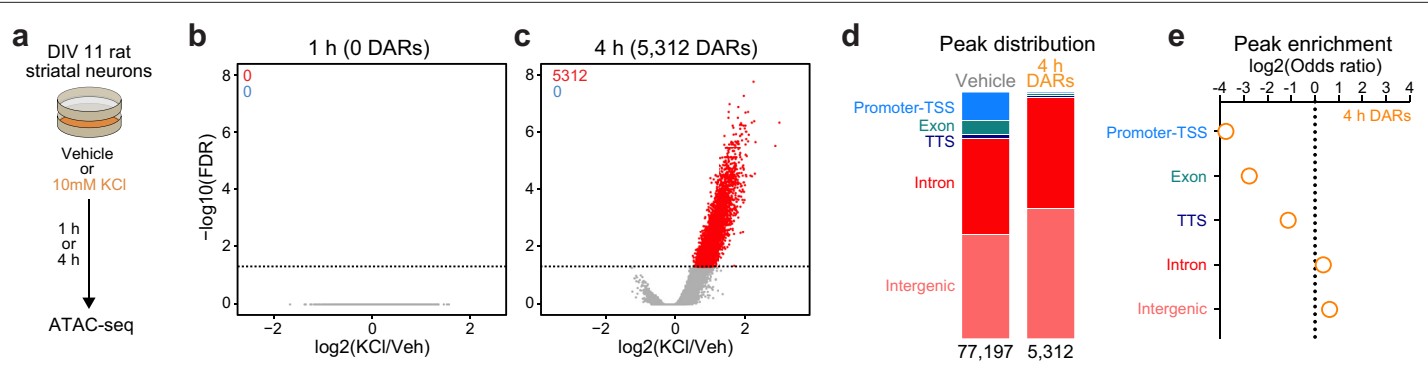

**Figure 2.** Activity-dependent chromatin remodeling in cultured primary rat striatal neurons. (**a**) DIV11 primary rat striatal neurons were treated with 10 mM KCl for 1 or 4 hr. Following treatment, ATAC-seq libraries were prepared. (**b, c**) Volcano plots displaying differentially accessible regions (DARs) after 1 and 4 hr of neuronal depolarization. (**d**) Genomic location of vehicle and 4 hr DAR ATAC peaks. (**e**) Odds ratio for genomic annotations of 4 hr DARs.

The online version of this article includes the following figure supplement(s) for figure 2:

**Figure supplement 1.** Transcription factor binding and histone modifications in 4 hr differentially accessible regions (DARs), random regions, and vehicle peaks.

factor activity (*Supplementary file 2*). However, LRG molecular function GO terms were distinct and included 'voltage gated channel activity' and 'G-protein-coupled receptor binding' (*Supplementary file 2*). Together, these results demonstrate that striatal neuron depolarization induces temporally and functionally distinct gene expression programs that can be classified as IEGs and LRGs.

Given that most LRGs emerged only after 4 hr of KCl depolarization, we next sought to determine whether this was due to the length of the depolarization stimulus or the time since stimulus onset. Striatal neurons were again treated with 10 mM KCl for 1 hr, followed by either a media wash off (replacement with conditioned media) or no wash off for 3 hr (*Figure 1—figure supplement 1d*). Reverse transcription-quantitative polymerase chain reaction (RT-qPCR) for a representative IEG (*Fos*) and a representative LRG (*Pdyn*) revealed that while *Fos* mRNA began to return to baseline in the wash off (1 hr stimulation) condition, *Pdyn* mRNA was equally elevated in response to 1 hr KCl (followed by 3 hr wash off) and 4 hr KCl stimuli (*Figure 1—figure supplement 1e, f*). Additionally, we found that both *Fos* and *Pdyn* expression were significantly elevated by 4 hr treatment with a variety of other stimuli (*Figure 1—figure supplement 1g, h*), including brain-derived neurotrophic factor (BDNF) and forskolin (FSK, an adenylyl cyclase activator). Moreover, the overall level of *Pdyn* mRNA was correlated with the level of *Fos* mRNA across individual replicates (*Figure 1—figure supplement 1i*), suggesting a direct link between IEGs and LRGs.

## Activity-dependent chromatin remodeling primarily occurs in non-coding genomic regions

In non-neuronal systems, LRG induction is dependent on activity-dependent chromatin remodeling at genomic enhancers (*Vierbuchen et al., 2017*). To identify the potential mechanisms governing transcriptomic reorganization following 4 hr of depolarization, we first sought to identify whether activity-dependent chromatin remodeling occurs in striatal neurons. Cultured rat embryonic striatal neurons were treated with 10 mM KCl for 1 or 4 hr and assay for transposase accessible chromatin followed by next-generation sequencing (ATAC-seq) libraries were prepared (*Figure 2a*). Strikingly, 1 hr of depolarization did not induce any chromatin remodeling that passed genome-wide cutoffs for statistical significance (*Figure 2b*). However, 4 hr of depolarization-induced genome-wide chromatin remodeling with 5312 differentially accessible regions (DARs, defined as regions with adjusted p-value <0.05), all of which become more open with depolarization (*Figure 2c*; *Supplementary file 3*). Previous studies have suggested that activity-dependent chromatin remodeling and other epigenetic processes occur in non-coding regions associated with genomic enhancers (*Vierbuchen et al., 2017*; *Feng et al., 2014*; *Sciumè et al., 2020*). To understand whether activity-dependent chromatin remodeling preferentially occurred in non-coding genomic regions, we calculated the odds ratio of

enrichment for all 4 hr DARs in comparison to baseline peaks identified in vehicle treated samples that did not overlap 4 hr DARs (termed 'vehicle' peaks; *Figure 2d*). Activity-regulated DARs were depleted in coding regions and promoters, but were enriched in intergenic and intronic regions of the genome (*Figure 2e*). This distribution is consistent with a function for these DARs as distal cis-regulatory enhancer elements.

Enhancers are marked by distinctive histone modifications, including histone acetylation (at H3K27) and increased ratios of histone monomethylation to trimethylation at H3K4 (H3K4me1:H3K4me3). In contrast, while promoters are also marked by H3K27ac, they tend to exhibit a lower H3K4me1:H3K4me3 ratio (*Carullo and Day, 2019*; *Chen et al., 2019*). To investigate whether 4 hr DARs were enriched for enhancer-associated histone modifications, we next leveraged a recent study (*Yeh et al., 2023*) that profiled histone modification enrichment throughout the mouse striatum. While DARs and vehicle peaks were enriched for H3K27ac and H3K4me1 (*Figure 2—figure supplement 1a, b*), DARs had a significantly higher H3K4me1:H3K4me3 ratio than vehicle peaks (*Figure 2—figure supplement 1c, d*). The preferential enrichment of DARs in non-coding regions coupled with the presence of H3K27ac and increased ratios of H3K4me1:H3K4me3 suggests that activity-dependent chromatin remodeling occurs at genomic enhancers in striatal neurons.

## Transcription factor motifs associated with IEGs are significantly enriched in 4 hr DARs

The combination of RNA- and ATAC-seq results suggests a temporal decoupling between induction of IEG transcription factors and activity-dependent chromatin remodeling. IEG transcription factors were activated following 1 hr of depolarization (*Figure 1b*), but activity-dependent chromatin remodeling only occurred following 4 hr of depolarization (*Figure 2c*). Furthermore, IEG transcription factors are critical mediators of activity-dependent chromatin remodeling in neuronal (*Su et al., 2017*) and non-neuronal cells (*Vierbuchen et al., 2017*). Thus, we predicted that IEG transcription factor motifs would be enriched in 4 hr DARs.

To test this possibility, we searched for enriched motifs using a database of experimentally validated transcription factor-binding motifs with HOMER (*Heinz et al., 2010*). Additionally, because millions of IEG motifs are found throughout the genome, we compared the enrichment between 4 hr DARs and peaks found in vehicle-treated samples. HOMER uses randomly selected background regions to compare the enrichment of motifs within a user identified peak set. This allowed us to calculate a percent enrichment, or the percent of background sequences with the motif subtracted from percent of target sequences with the motif. While no transcription factor motifs exhibited enrichment greater than 50% in the vehicle peak dataset (*Figure 3a*), eight motifs exceed this threshold within 4 hr DARs (*Figure 3b*). Interestingly, all motifs correspond to versions of the consensus motif for the AP-1 family of activity-dependent transcription factors, with 93% of 4 hr DARs containing an AP-1 motif (*Figure 3c*). In addition to calculation of a percent enrichment, we calculated the enrichment of specific motifs across DARs and vehicle peaks. AP-1, as well as its subunits FOS, FOSL2, and JUNB, were significantly enriched at the center of DARs and not in vehicle peaks (*Figure 3d*). Recently, binding sites for the AP-1 family member ΔFosB were assayed in the adult mouse nucleus accumbens using CUT&RUN (*Yeh et al., 2023*). Binding sites identified in this study were then mapped to the rat genome. ΔFosB is significantly enriched in DARs, but not in vehicle peaks or a set of 5312 random regions that are the same size as DARs (*Figure 2—figure supplement 1e*). Additional analyses identified that MEF2C motifs were enriched at the center of DARs (*Figure 3c, d*). MEF2C is a member of the MEF2 family of proteins that are integral regulators of synaptic plasticity in the developing brain and interact with histone deacetylases to alter chromatin accessibility (*Zhang et al., 2016*; *Shalizi and Bonni, 2005*; *Dietrich et al., 2012*). ISL1, is an integral regulator of MSN development (*Ehrman et al., 2013*), was also enriched at the center of DARs, with 95% of DARs containing an ISL1 motif (*Figure 3c, d*).

The HOMER database allowed us to investigate the enrichment of over 400 transcription factor motifs in over 5300 DARs. To understand if DARs could be separated based on the transcription factor motifs present within these regions, we used uniform manifold approximation and projection (UMAP; a dimensionality reduction technique) (*McInnes et al., 2018*) and density-based clustering (*Hahsler et al., 2019*) to separate DARs based on the presence, absence, or count of annotated transcription factor motifs. This analysis identified three major clusters with some DARs labeled as outliers (cluster

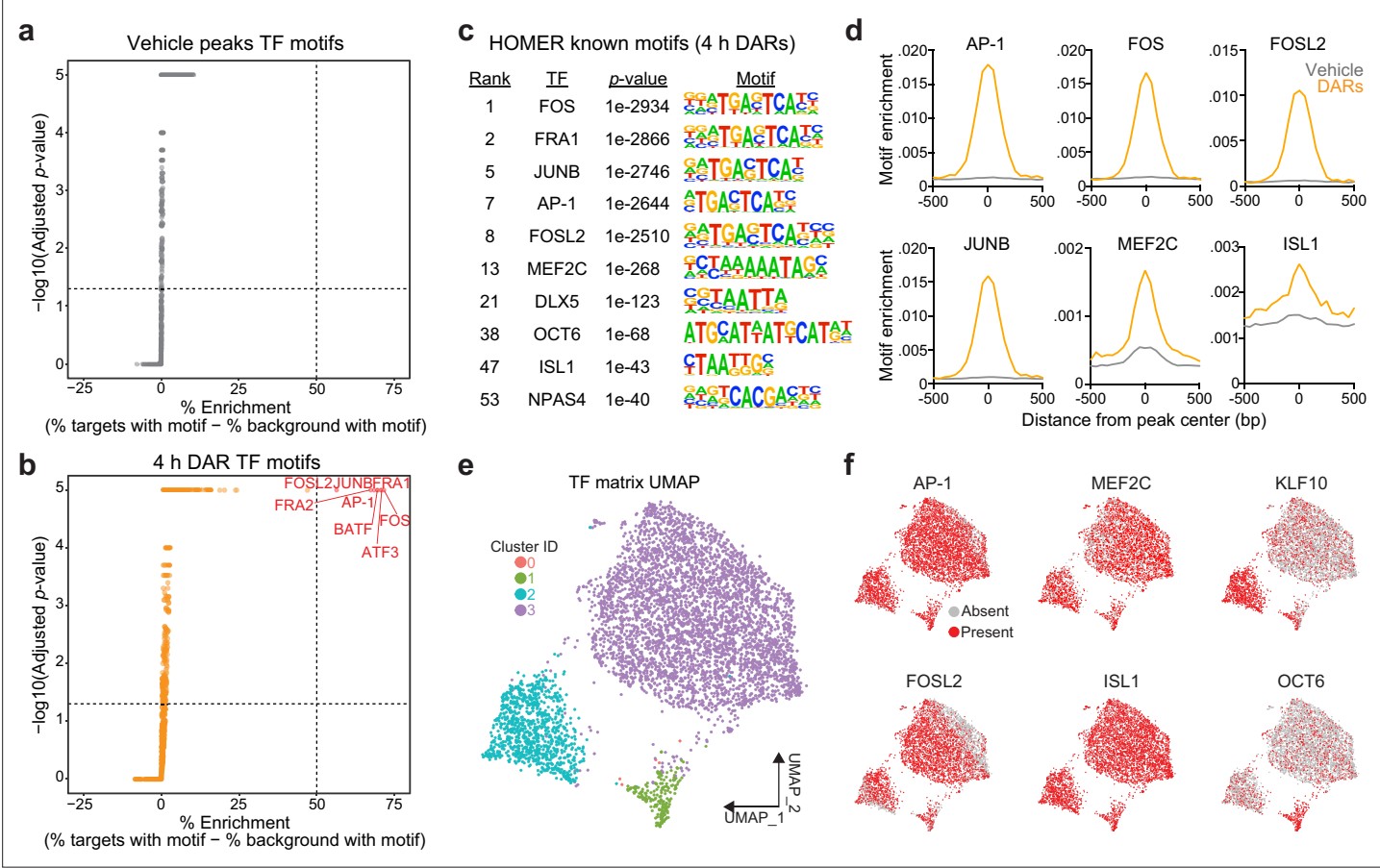

**Figure 3.** Motifs for activity-dependent transcription factors are significantly enriched in 4 hr differentially accessible regions (DARs). (**a, b**) Plots showing enrichment of specific transcription factor motifs in vehicle peaks and 4 hr DARs. Motifs with significant adjusted p-values and a percent enrichment greater than 50% are shown in red and labeled with the corresponding transcription factor. (**c**) Representative results from HOMER known motif enrichment analysis conducted using 4 hr DAR peak set. (**d**) Motifs for activity-dependent transcription factors (TFs) (AP1, FOS, FOSL2, and JUNB), MEF2C, and ISL1 are enriched at the center of 4 hr DARs but not baseline (vehicle) peaks. Motif histogram distribution is represented as motifs/bp/peak. (**e**) Uniform manifold approximation and projection (UMAP) generated using transcription factor motif enrichment within the 4 hr DARs. (**f**) UMAPs colored by the presence of absence of specific transcription factor motifs. KLF10 and OCT6 specifically mark clusters 2 and 1, respectively.

0) (*Figure 3e*). Unsurprisingly, clusters were not defined by the presence of AP-1, FOSL2, MEF2C, or ISL1, transcription factors that are significantly enriched at the center of DARs (*Figure 3f*). However, cluster 2 was marked by the presence of the KLF10 motif and cluster 1 was marked by the presence of a motif corresponding to OCT6 (*Figure 3f*). Taken together, these results demonstrate a significant enrichment of IEG motifs within 4 hr DARs and suggest that IEGs may participate in activity-dependent chromatin remodeling in striatal neurons.

## De novo protein translation is required for activity-dependent chromatin remodeling

The significant enrichment of IEG transcription factor motifs within 4 hr DARs suggests that these inducible transcription factors may be required for subsequent activity-dependent chromatin remodeling. Furthermore, the temporal decoupling between IEG expression and activity-dependent chromatin remodeling suggests that IEGs must be translated before acting on genomic regions to induce an open chromatin state. To test the hypothesis that de novo protein translation is required for activity-dependent chromatin remodeling, we repeated ATAC-seq after 4 hr of depolarization in combination with protein synthesis inhibition. Cultured rat embryonic striatal neurons were pretreated with 40 µM anisomycin, a translation inhibitor, for 30 min followed by 10 mM KCl for 4 hr (*Figure 4a*). As expected, 40 µM anisomycin was sufficient to block depolarization-induced translation of FOS

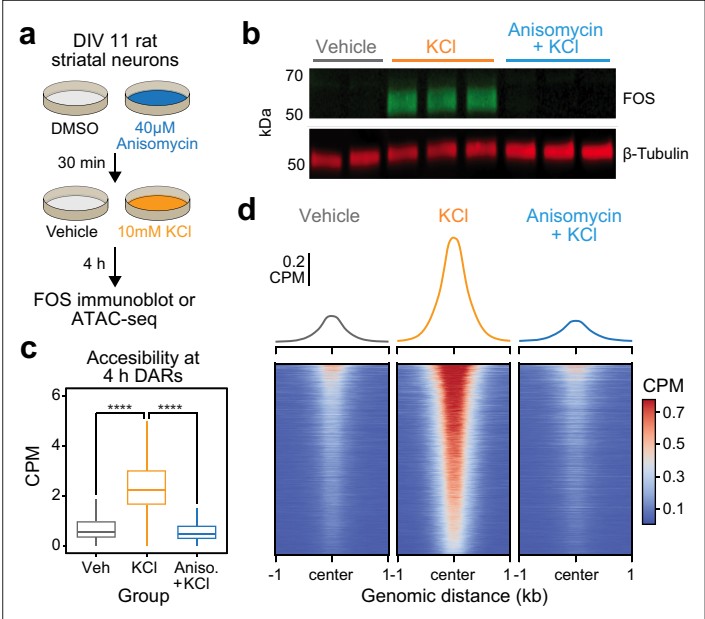

**Figure 4.** Activity-dependent chromatin remodeling requires protein translation. (**a**) Experimental design. DIV11 primary rat striatal neurons were treated with Dimethyl Sulfoxide (DMSO) or anisomycin for 30 min followed by 4 hr of depolarization with 10 mM KCl. (**b**) Western blot for FOS and β-tubulin for cells treated with vehicle, 10 mM KCl, or 10 mM KCl + 40 µM anisomycin. (**c**) Boxplots demonstrating the effects of anisomycin on activity-dependent chromatin remodeling. One-way analysis of variance (ANOVA) with Tukey's multiple comparisons test, ****$p < 0.0001$. (**d**) Heatmaps and mean accessibility plots from 4 hr differentially accessible regions (DARs). For heatmaps, each row represents a single DAR. CPM = counts per million.

The online version of this article includes the following figure supplement(s) for figure 4:

**Figure supplement 1.** Enhancers for *Fos* are open at baseline and do not undergo activity-dependent chromatin remodeling.

protein, as determined by immunoblotting (*Figure 4b*). Targeted analysis of the previously identified 5312 DARs demonstrated replication of activity-dependent chromatin remodeling at these regions (*Figure 4c*). Pretreatment with anisomycin completely blocked activity-dependent chromatin remodeling across the genome (*Figure 4c, d*), demonstrating that de novo protein translation is required for activity-dependent changes in chromatin accessibility. Furthermore, this result suggests that chromatin remodeling induced by neuronal depolarization is regulated by IEG transcription factors.

## Activity-dependent *Pdyn* transcription is regulated by IEGs and protein translation

Given that activity-dependent chromatin remodeling across the genome required protein translation, we next sought to identify discrete regions near LRGs that may serve as activity-dependent enhancers. We predicted that regions serving as activity-dependent enhancers for LRGs would be: (1) located in non-coding regions of the genome, (2) inaccessible at baseline and accessible following depolarization, and (3) inaccessible when depolarization was paired with protein synthesis inhibition. These regions would be fundamentally different from enhancers regulating IEGs (such as the enhancers within the *Fos* locus), which are accessible at baseline and do not undergo activity-dependent chromatin remodeling (*Figure 4—figure supplement 1*). We identified a putative enhancer ~45 kb upstream of the *Pdyn* TSS that met all of these criteria (*Figure 5a*). While *Pdyn* is an LRG that plays important roles in striatal function, the *Pdyn* promoter does not undergo activity-dependent chromatin remodeling (*Figure 5a*). Next, we predicted that if this differently accessible chromatin region in the *Pdyn* locus serves as enhancer for *Pdyn*, then blocking protein translation should also attenuate activity-dependent *Pdyn* transcription. In support of this prediction, anisomycin pretreatment completely blocked activity-dependent *Pdyn* transcription, as detected with RT-qPCR (*Figure 5b*). Additionally, anisomycin pretreatment attenuated baseline *Pdyn* mRNA levels in the absence of

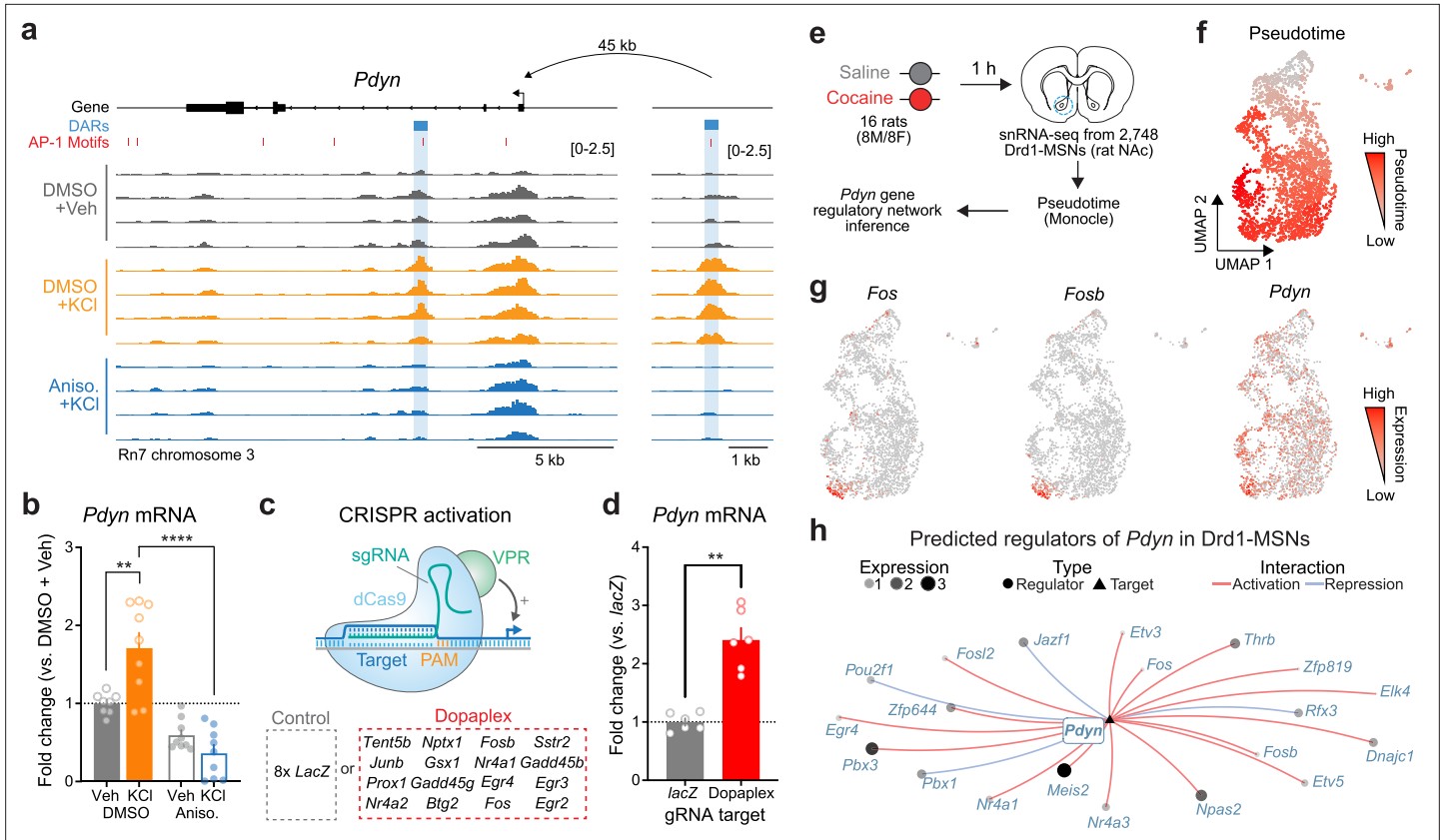

**Figure 5.** Transcriptional regulation of *Pdyn* mRNA. (**a**) ATAC-seq tracks at the *Pdyn* gene locus of embryonic striatal neurons treated with DMSO + Vehicle, DMSO + KCl, or anisomycin + KCl. A differentially accessible region (DAR) 45 kb upstream of the *Pdyn* TSS in a non-coding region becomes accessible with depolarization only with intact protein translation. (**b**) RT-qPCR for *Pdyn* mRNA from DIV 11 rat striatal neurons treated with vehicle, KCl, anisomycin, or anisomycin + KCl. Induction of *Pdyn* mRNA by KCl is blocked by anisomycin pretreatment (one-way analysis of variance [ANOVA] with n=8-9 per group with Tukey's multiple comparisons test **p < 0.01, ****p < 0.0001). Data expressed as mean + SEM. (**c**) Targeted activation of dopamine-regulated immediate early genes (IEGs) with CRISPR activation (data from *Savell et al., 2020*). dCas9-VPR was transduced with multiplexed sgRNAs targeting 16 IEGs. (**d**) *Pdyn* mRNA is upregulated following CRISPR-based activation of 16 IEGs. Mann–Whitney test with n=6 per group. **p < 0.01. Data expressed as mean + SEM. (**e**) Pseudotime analysis to predict regulators of *Pdyn* expression in Drd1-MSNs was performed with available snRNA-seq data from the rat nucleus accumbens (*Savell et al., 2020*). (**f, g**) Feature plots for Pseudotime, *Fos*, *Fosb*, and *Pdyn* in Drd1-MSNs. For these feature plots, a brighter red color is associated with a higher pseudotime and gene expression. (**h**) Gene regulatory network reconstruction from single-cell trajectories identifies predicted regulators of *Pdyn* in Drd1-MSNs.

The online version of this article includes the following source data and figure supplement(s) for figure 5:

**Source data 1.**

**Figure supplement 1.** Predicted regulators of *Pdyn* and *Scg2* in Drd1-MSNs.

stimulation, suggesting that basal *Pdyn* expression is comprised of both constitutive and activity-dependent transcriptional events (***Figure 5b***).

To determine whether *Pdyn* is directly regulated by IEG transcription factors, we leveraged a previously published RNA-seq dataset that used CRISPR activation tools to overexpress 16 IEGs that are upregulated in striatal neurons following dopamine stimulation (***Savell et al., 2020***; ***Figure 5c***). CRISPR-based activation of 16 IEGs (including *Fos*, *Fosb*, and *Junb*) resulted in significant upregulation of *Pdyn* mRNA (***Figure 5d***), demonstrating that activation of IEGs is sufficient to upregulate *Pdyn* expression. To further investigate the transcriptional regulation of *Pdyn*, we leveraged a publicly available single-nucleus RNA-sequencing dataset from rats treated with a single dose of cocaine (or saline as a control; ***Figure 5e***; ***Savell et al., 2020***). Because these rats were sacrificed 1 hr after injection, we reasoned that LRGs have not yet been fully activated by cocaine experience at this timepoint. However, using pseudotime analysis to reconstruct gene regulatory networks allowed us to identify predicted upstream regulators of *Pdyn* transcription. In prior work, we demonstrated that distinct

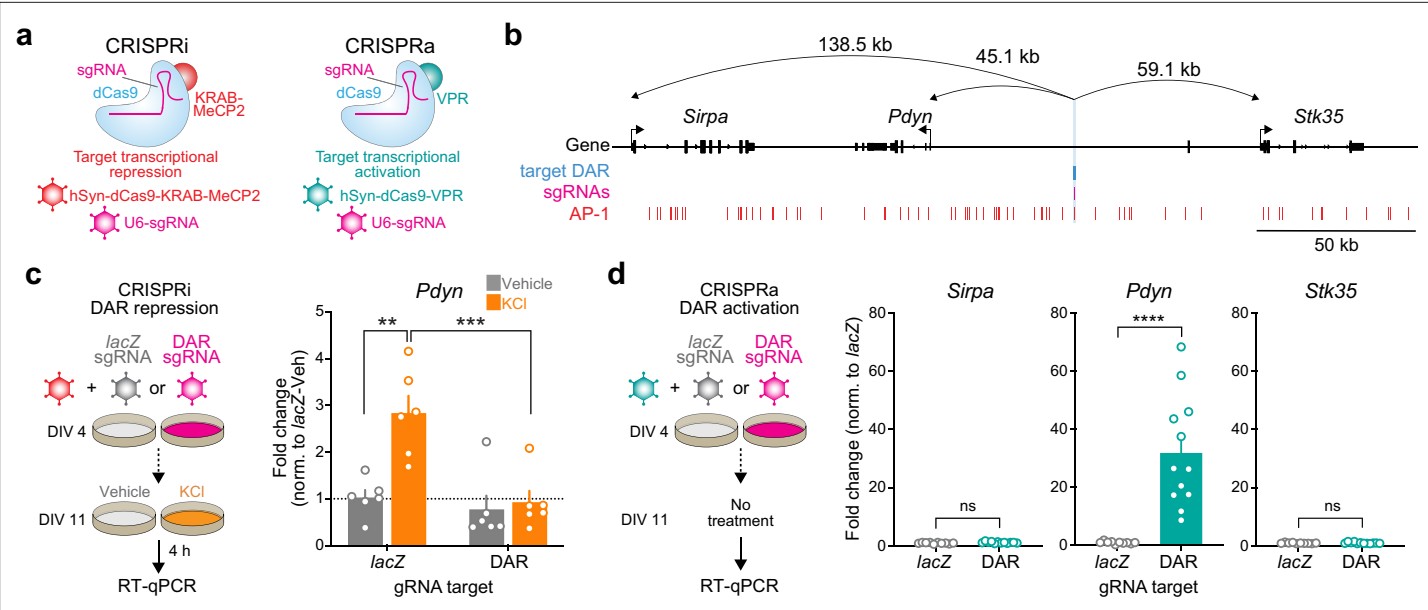

**Figure 6.** CRISPR-based functional validation of a novel *Pdyn* enhancer. (**a**) Viral strategy for functional validation of putative enhancers using CRISPR interference with dCas9-KRAB-MeCP2 and CRISPR activation with dCas9-VPR. (**b**) CRISPR sgRNAs (4×) were designed to target the activity-regulated differentially accessible region (DAR) 45.1 kb upstream of *Pdyn* in the rat genome. (**c**) CRISPRi at the *Pdyn* DAR blocks activity-dependent induction of *Pdyn* mRNA. Cultured embryonic striatal neurons were transduced with dCas9-KRAB-MeCP2 and sgRNAs targeting lacZ (non-targeting control) or the *Pdyn* DAR. Neurons were then treated with vehicle or 10 mM KCl for 4 hr prior to RT-qPCR. One-way analysis of variance (ANOVA) with n=6 per group and Tukey's multiple comparisons test \*\*p < 0.01, \*\*\*p < 0.001. Data expressed as mean + SEM. (**d**) CRISPRa at the *Pdyn* DAR selectively upregulates *Pdyn* mRNA without altering expression of the nearest upstream and downstream genes. Striatal neurons were transduced with dCas9-VPR and sgRNAs targeting lacZ or the *Pdyn* DAR. Following RNA extraction at DIV 11, RT-qPCR was used to measure expression of *Pdyn*, *Sirpa*, and *Stk35*. Mann–Whitney test with n=12 per group \*\*\*\*p < 0.0001. Data expressed as mean + SEM.

The online version of this article includes the following source data for figure 6:

**Source data 1.**

**Source data 2.**

populations of Drd1-MSNs have differential responses to cocaine (***Phillips et al., 2023***). Therefore, pseudotime trajectory graphs were constructed such that the highest pseudotime (cells colored bright red) marked cells that were transcriptionally activated by cocaine (i.e., expressed high levels of IEGs) (***Figure 5f, g***). *Pdyn* mRNA levels were also higher in cells with high *Fos* and *FosB* expression (***Figure 5g***). Finally, a gene regulatory network was constructed for cells with high IEG levels. This network predicted that the IEG transcription factors *Fos*, *Fosb*, *Fosl2*, *Nr4a1*, and *Nr4a2* all regulate *Pdyn* (***Figure 5h***). As a positive control, we also generated a gene regulatory network for *Scg2*, an LRG that was previously demonstrated to be regulated by *Fos* (***Yap et al., 2021***). This network also predicted *Fos* as a regulator of *Scg2* (***Figure 5—figure supplement 1***), demonstrating that this computational technique can replicate experimental findings.

## CRISPR-based functional validation of a novel *Pdyn* enhancer

The observation that *Pdyn* DAR accessibility and *Pdyn* mRNA were both activity- and translation-dependent suggests that this site acts as a stimulus-regulated enhancer for *Pdyn*. To test this hypothesis, we used a catalytically dead version of Cas9 (dCas9) fused to the transcriptional activator VPR (CRISPRa) or the transcriptional repressor KRAB-MeCP2 (CRISPRi) (***Duke et al., 2020***; ***Savell et al., 2019a***) to activate or silence this region on demand (***Figure 6a, b***). To test the necessity of the DAR in mediating activity-dependent *Pdyn* transcription, we used single-guide RNAs (sgRNAs) to target dCas9-KRAB-MeCP2 to the DAR in the presence of depolarization. As a negative control, we transduced cultured striatal neurons with a sgRNA for *lacZ*, a bacterial gene not found in the mammalian genome. As expected, activity-dependent *Pdyn* upregulation was observed in neurons transduced

with *lacZ* sgRNAs and dCas9-KRAB-MeCP2 (**Figure 6c**). However, KCl-induced *Pdyn* upregulation was blocked in neurons transduced with DAR-targeting sgRNAs and dCas9-KRAB-MeCP2 (**Figure 6c**). Additionally, baseline *Pdyn* mRNA levels were attenuated by CRISPRi of the *Pdyn* DAR (**Figure 6c**), suggesting that this enhancer may be accessible at baseline in some cells or as a consequence of spontaneous neuronal activity. These results demonstrate that the DAR upstream of the *Pdyn* TSS is necessary for activity-dependent *Pdyn* transcription.

Next, we transduced neurons with the same sgRNAs targeting *lacZ* or the *Pdyn* DAR and dCas9-VPR (CRISPRa) to test if activation of this DAR is sufficient to promote *Pdyn* transcription. *Pdyn* mRNA levels were significantly increased in neurons transduced with DAR-targeting sgRNAs (**Figure 6d**), demonstrating that transcriptional activation of the DAR is sufficient for upregulation of *Pdyn* mRNA levels. To test the specificity of the DAR for regulation of *Pdyn*, we also measured mRNA levels of the two closest genes within the *Pdyn* locus, *Sirpa* and *Stk35*. CRISPRa-based transcriptional activation of the DAR did not result in significant upregulation of *Sirpa* or *Stk35*. Taken together, these CRISPR-based functional assays demonstrate that the DAR upstream of the *Pdyn* TSS is a genomic enhancer that is necessary, sufficient, and specific for *Pdyn* transcription.

## *PDYN* is a cell type-specific LRG in the human genome regulated by a conserved genomic enhancer

Because genomic enhancers are critical regulators of genes important for shared biological functions, many of these elements are conserved across species. Furthermore, regions undergoing activity-dependent chromatin remodeling in human neurons harbor genetic variants associated with the development of neuropsychiatric disorders (**Sanchez-Priego et al., 2022**). Thus, we next set out to understand if *PDYN* is an LRG in the human genome, and whether the identified rat enhancer is conserved. To test whether *PDYN* is an LRG in the human genome, we leveraged publicly available RNA-seq datasets from cultured human GABAergic neurons (**Sanchez-Priego et al., 2022**). These neurons were depolarized for 0.75 or 4 hr, timepoints that allow for the investigation of IEGs and LRGs, respectively. Like cultured rat embryonic striatal neurons, *PDYN* is only upregulated following 4 hr of depolarization (**Figure 7a–c**). This study also performed the same experiments in cultured human glutamatergic neurons, allowing us to investigate whether *PDYN* is a cell type-specific LRG. Analysis of *PDYN* at several timepoints in both glutamatergic and GABAergic neurons demonstrated that *PDYN* is only upregulated in GABAergic neurons following 4 hr of depolarization (**Figure 7d**).

This published study also performed ATAC-seq on human glutamatergic and GABAergic neurons following 0 and 90 min of depolarization. While 90 min does not provide the same temporal resolution for chromatin remodeling at genomic enhancers as our experiments in rat neurons, it is a time-point in which chromatin remodeling may initially occur following stimulation. Interestingly, the rat *Pdyn* DAR is conserved in a region that is also upstream of the human *PDYN* TSS (**Figure 7e**), and this region undergoes time-dependent increases in chromatin accessibility following KCl stimulation along with other identified DARs in this locus (**Figure 7f**). Furthermore, activity-induced chromatin remodeling at the conserved and experimentally validated DAR only occurs in human GABAergic neurons (**Figure 7—figure supplement 1**). To test if this enhancer is sufficient for upregulation of *PDYN* transcription in human cells, we transfected HEK-293T cells with plasmids to express dCas9-VPR machinery and sgRNAs targeting the conserved region (**Figure 7g**). Transcriptional activation of the conserved DAR was sufficient to upregulate human *PDYN* transcription (**Figure 7h**). Together, these results suggest that *PDYN* is a cell type-specific LRG within the human genome that is regulated by a conserved enhancer element.

## *Pdyn* enhancer is accessible in the adult rat striatum in a cell type-specific manner

To determine whether the activity-dependent *Pdyn* enhancer characterized in vitro is also functional in the adult brain, we performed snATAC-seq using nuclei from male and female adult Sprague-Dawley rats that received once daily intraperitoneal cocaine injections for 7 consecutive days (repeated cocaine; **Figure 8a**). 10,085 snATAC-seq nuclei were sequenced and integrated with a previously published snRNA-seq dataset (**Phillips et al., 2023**) that contained NAc nuclei from rats treated with repeated cocaine injections to identify cell types. Nuclei were distributed across 14 distinct cell types (**Figure 8b**) that include previously identified neuronal and non-neuronal cell populations (**Savell et al.,**

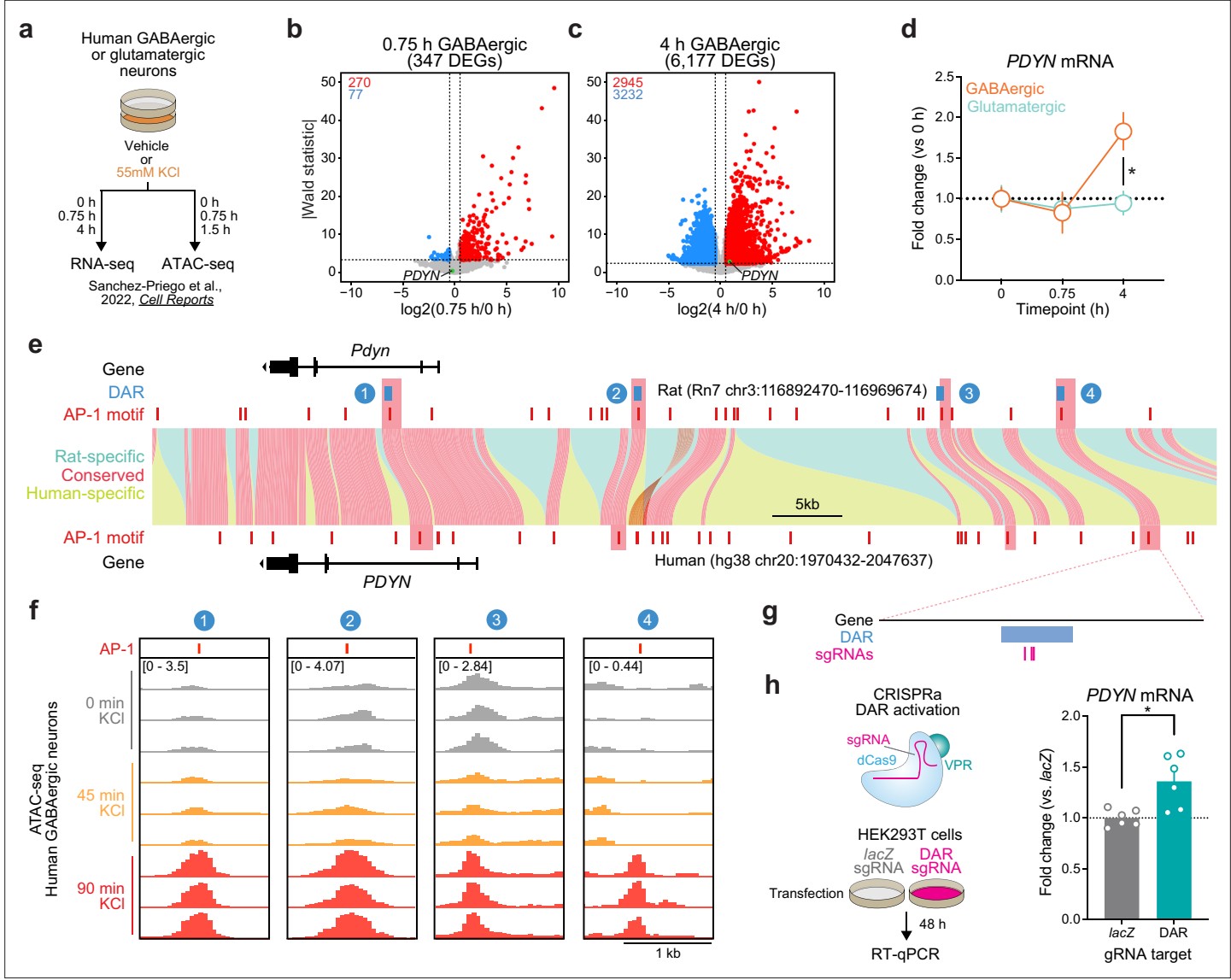

**Figure 7.** Identification and validation of a conserved *PDYN* enhancer in the human genome. (**a**) Experimental design for published RNA- and ATAC-seq datasets from human GABAergic and glutamatergic neurons treated with 55 mM KCl. (**b, c**) Volcano plots for human GABAergic neurons treated with 55 mM KCl for 0.75 or 4 hr. *PDYN* is a significant differentially expressed gene (DEG) at 4 hr, but not 0.75 hr. (**d**) *PDYN* is significantly upregulated 4 hr after a KCl stimulus, but only in GABAergic neurons. Two-way analysis of variance (ANOVA) with Tukey's multiple comparison's test. *p < 0.05. (**e**) Linear synteny view of the rat and human *Pdyn/PDYN* locus reveals shared conservation of four distinct activity-dependent differentially accessible regions (DARs) identified in rat striatal neurons. (**f**) ATAC-seq tracks of GABAergic neurons treated with 55 mM KCl for 0, 0.75, or 1.5 hr at the human *PDYN* locus. Regions conserved between rats and humans undergo dynamic remodeling in human GABAergic cells at 1.5 hr after stimulation. A region homologous to the rat *Pdyn* enhancer characterized in *Figure 6* is 63.7 kb upstream of the human *PDYN* gene. (**g**) Location of CRISPR sgRNAs in the human genome for CRISPR-based activation of the *PDYN* DAR in human cells. (**h**) CRISPRa at the human *PDYN* DAR is sufficient to upregulate *PDYN* mRNA in HEK293T cells. Mann–Whitney test. *p < 0.05.

The online version of this article includes the following source data and figure supplement(s) for figure 7:

**Source data 1.**

**Figure supplement 1.** Conserved activity-regulated *PDYN* differentially accessible region (DAR) is selective for GABAergic neurons.

*2020*; *Phillips et al., 2023*). Visualization and accessibility peak-calling identified that the validated *Pdyn* enhancer locus was accessible in both Drd1- and Grm8-MSNs (*Figure 8b*), populations that also express high levels of *Pdyn* at the mRNA level. Furthermore, co-accessibility analysis identified that accessibility of the *Pdyn* enhancer was correlated with accessibility of the *Pdyn* promoter across individual cells (*Figure 8c*). This result demonstrates that the region is accessible, and coaccessible with

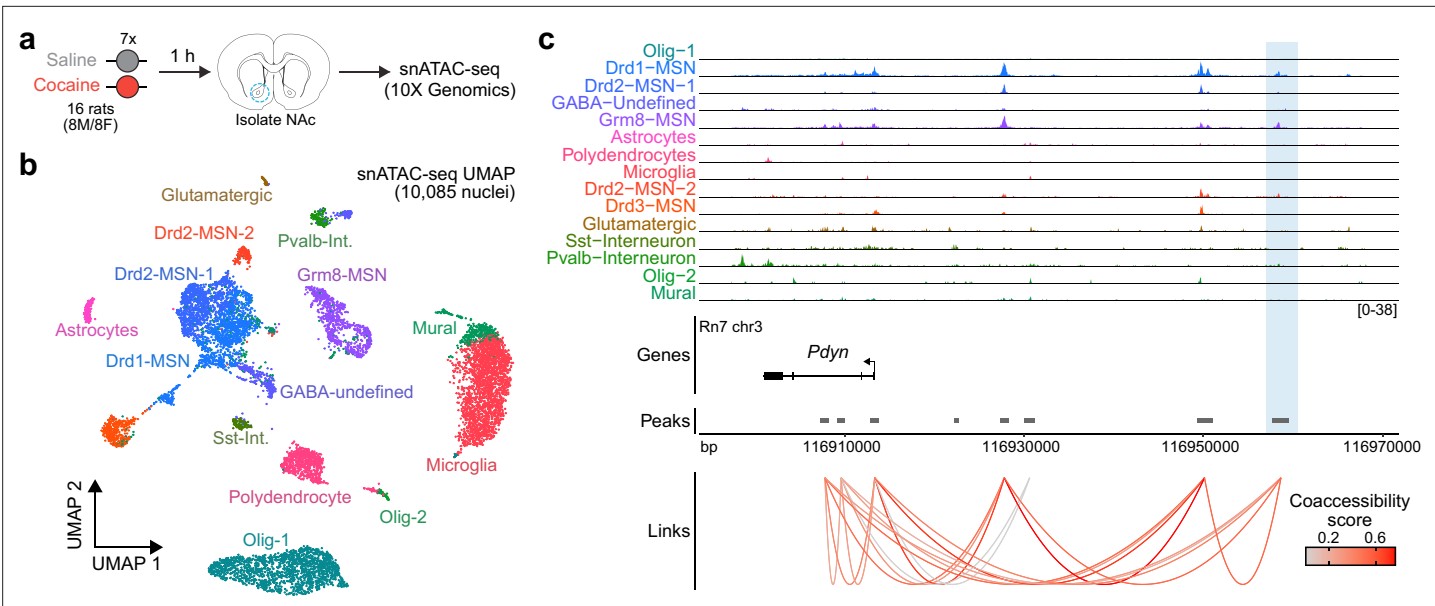

**Figure 8.** Characterization of the rat *Pdyn* differentially accessible region (DAR) in the adult nucleus accumbens at single-cell resolution. (**a**) Experimental design. (**b**) Uniform manifold approximation and projection (UMAP) of 10,085 nuclei from adult rat nucleus accumbens (NAc). (**c**) Genome tracks displaying snATAC-seq data at the *Pdyn* locus. The experimentally validated *Pdyn* DAR (highlighted in blue) exhibits high co-accessibility with the proximal promoter for *Pdyn* as well as other nearby accessible regions.

the *Pdyn* promoter, in vivo. Furthermore, this dataset suggests that the *Pdyn* enhancer is functional in the adult rat brain in selected cell populations that also express *Pdyn* mRNA.

## Discussion

Here, we used transcriptomic and epigenomic profiling to characterize activity-dependent transcriptional and epigenomic waves in cultured embryonic rat striatal neurons, an in vitro model relevant for studying striatal function and neuropsychiatric disease. These experiments characterized a well-studied IEG expression program, which consisted primarily of activity-dependent transcription factors, as well as a delayed LRG expression program (*Figure 1a–c*). While IEGs are required for cellular and behavioral adaptations, they control this process through the transcriptional regulation of LRGs. LRGs are both functionally and temporally distinct from IEGs. IEGs primarily encode transcription factors and co-activators, while LRGs encode opioid peptides, transporters, and other proteins involved in synaptic plasticity (*Yap and Greenberg, 2018*; *Tyssowski et al., 2018*). For example, FOS regulates transcriptional activation of the LRG *Scg2* (*Yap et al., 2021*), a gene that encodes several neuropeptides that regulate inhibitory plasticity (*Iwase et al., 2014*). IEGs, such as *Fos*, may regulate LRGs by their involvement in activity-dependent chromatin remodeling, as *Fos* mRNA induction is required for activity-dependent chromatin remodeling in dentate gyrus following neuronal stimulation (*Su et al., 2017*). Our results highlight how this process modulates the expression of prodyorphin, a neuropeptide with critical functions in the striatum. Furthermore, our work defines the dynamic nature of activity-dependent chromatin accessibility changes in striatal neurons using comprehensive approaches.

Surprisingly, analysis of both transcriptomic and epigenomic data revealed a temporal decoupling between transcriptional activation of IEGs and chromatin remodeling. While 1 hr of depolarization was sufficient to induce hundreds of transcriptional changes, genome-wide chromatin remodeling was only observed following 4 hr of depolarization. This result contrasts with previously published studies that observed genome-wide chromatin remodeling in the mouse dentate gyrus following 1 hr of electroconvulsive stimulation (*Su et al., 2017*), or in hippocampal excitatory neurons following seizure induction with kainic acid (*Fernandez-Albert et al., 2019*). Differences in the temporal dynamics of activity dependent chromatin remodeling could be due to differences in both the cell type of interest (hippocampal vs. striatal neurons), as well as the type of stimulation (e.g., electroconvulsive

stimulation vs. depolarization). Nevertheless, our studies agree in that a significant proportion of regions undergoing remodeling become more accessible following stimulation (*Figure 2c*). While it is known that IEGs engage in activity-dependent chromatin remodeling at LRGs in other brain regions and cell types, this process has seldom been studied in an addiction-relevant cell type such as striatal neurons. Additionally, drugs of abuse engage IEGs in the striatum (*Savell et al., 2020*; *Phillips et al., 2023*; *Bertran-Gonzalez et al., 2008*; *Guez-Barber et al., 2011*; *Hope et al., 1994*; *Kelz et al., 1999*; *Graybiel et al., 1990*; *Moratalla et al., 1993*), and interference with IEG induction prevents subsequent cellular and behavioral adaptations caused by psychoactive drugs (*Kelz et al., 1999*; *Zhang et al., 2006*; *Carlezon et al., 1998*). Thus, investigation of IEG-dependent chromatin remodeling at enhancers regulating addiction-relevant LRGs is important for understanding how rapid gene expression changes might result in persistent cellular and behavioral adaptations.

The accepted model of activity-dependent transcription posits that LRG activation is dependent on IEGs. To this point, blocking *Fos* mRNA induction via shRNA in the dentate gyrus is sufficient to significantly attenuate activity-dependent chromatin remodeling (*Su et al., 2017*). These data demonstrate that *Fos* is required for some level of chromatin remodeling in the brain but does not provide evidence regarding the mechanisms through which it is involved. In non-neuronal cells, AP-1 acts as a pioneer factor at genomic enhancers by guiding the SWI/SNF chromatin remodeling complex to target regions (*Vierbuchen et al., 2017*; *Wolf et al., 2023*). Our data suggest a similar mechanism may regulate activity-dependent chromatin remodeling at genomic enhancers in neurons. First, AP-1 motifs and ΔFosB-binding sites are significantly enriched within 4 hr DARs (*Figure 3b–d*, *Figure 2— figure supplement 1e*). Second, blocking translation of IEGs, which include a significant number of AP-1 family members, completely blocks activity-dependent chromatin remodeling (*Figure 4c, d*). Thus, we speculatively propose a model in which neuronal stimulation results in the transcription and translation of AP-1 family members. These AP-1 family members then interact with the SWI/SNF complex to induce chromatin remodeling at genomic enhancers. However, while AP-1 is thought to aid in enhancer selection, it is expressed in a non-cell type-specific manner and must choose from over 1 million potential AP-1 motifs in the genome. Thus, additional factors must be present to ensure precise cell type-specific responses to activity. The use of the HOMER database allowed us to explore the enrichment of over 400 additional transcription factor motifs, including ISL1, an MSN-selective transcription factor. ISL1 is significantly enriched in DARs (*Figure 3c, d*). The combined enrichment of AP-1 and ISL1 at 4 hr DARs suggests that cell type-specific transcription factors may be working in conjunction with activity-dependent transcription factors to induce chromatin remodeling. We envision three possible mechanisms through which cell type-specific transcription factors may be involved in this process. First, population-specific transcription factors may be directly interacting with AP-1 or chromatin remodeling complexes to induce remodeling at these sites. Second, ISL1 may stochastically bind these regions, resulting in specific temporal windows in which AP-1 might bind in a cooperative fashion. Finally, because cell type-specific transcription factors are often activated during development, they may epigenetically alter potential binding sites proximal to AP-1 motifs, resulting in a preferential targeting of these AP-1 motifs at later timepoints. Any of these mechanisms would allow distinct cell types to induce specific LRG programs, even with homogenous IEG activation, and ultimately enable the same stimulus to produce different cellular adaptations in different cell types.

Multiple studies have demonstrated that most regions undergoing activity-dependent chromatin remodeling are genomic enhancers (*Vierbuchen et al., 2017*; *Malik et al., 2014*). Analysis of 4 hr DARs from this study identified a significant enrichment for DARs in non-coding regions and a depletion in coding regions (*Figure 2d, e*), suggesting that activity-dependent chromatin remodeling in striatal neurons is also occurring at genomic enhancers. These data also demonstrate a distinction between the chromatin profiles of IEG and LRG enhancers. IEG enhancers are 'poised' at baseline, while LRG enhancers are largely inaccessible in without stimulation. For example, enhancers within the *Fos* locus are accessible in both vehicle- and KCl-treated samples (*Figure 4—figure supplement 1*), while the *Pdyn* enhancer is only accessible with neuronal depolarization (*Figure 5a*). In addition, LRG enhancers contain motifs for activity-dependent transcription factors (*Figure 3a–e*) and are dependent on de novo protein translation (*Figure 4c, d*). While LRG enhancers become accessible only with activity, both IEG and LRG enhancers are in non-coding regions of the genome and are marked by specific histone modifications like H3K27ac and H3K4me1 (*Malik et al., 2014*; *Carullo and Day, 2019*; *Zentner et al., 2011*; *Carullo et al., 2020b*; *Figure 2—figure supplement 1*).

While the enrichment of DARs in putative enhancer elements is intriguing, we also wanted to understand whether enhancers regulate nearby LRGs. To do this, we identified a DAR upstream of the *Pdyn* TSS that may serve as a functional enhancer. This region is also accessible in the adult rat NAc (*Figure 8c*). Furthermore, this region is co-accessible with the *Pdyn* promoter and has the highest level of accessibility in Drd1- and Grm8-MSNs, suggesting that the enhancer is functional in vivo and may exhibit some level of cell type specificity (*Figure 8c*). However, a larger dataset containing more nuclei will be needed to test for any drug-specific chromatin remodeling at this region. CRISPR-based functional assays demonstrated that the DAR upstream of the *Pdyn* TSS serves as a genomic enhancer that is necessary, sufficient, and specific for transcription of *Pdyn*. We were particularly interested in *Pdyn* because it encodes the dynorphin neuropeptides that are agonists for the kappa opioid receptor (*Chavkin et al., 1982*). The kappa opioid receptor system has been the target of several antagonists that reduce drug-taking behaviors in pre-clinical models (*Prisinzano et al., 2005*; *Zamarripa et al., 2020*; *Chavkin, 2011*; *Valenza et al., 2020*). Identification of a novel genomic enhancer for *Pdyn* would represent a novel therapeutic target. Furthermore, *Pdyn* is regulated by dopamine (*Berke et al., 1998*) and drugs of abuse (*Carlezon et al., 1998*; *Cole et al., 1995*; *Jenab et al., 2002*; *Sun et al., 2020*; *Corchero et al., 1997*; *Piechota et al., 2012*), and polymorphisms and structural variants within the human *PDYN* gene locus are associated with drug abuse (*Yuferov et al., 2009*; *Clarke et al., 2012*). In particular, one polymorphism affects an AP-1-binding site within the *PDYN* gene promoter, which may result in less *PDYN* transcription and an increased risk for cocaine dependence (*Yuferov et al., 2009*). Thus, the identification of a functional, conserved enhancer for *PDYN* in the human genome provides a novel potential therapeutic target capable of regulating stimulus-dependent changes in *PDYN* expression while leaving constitutive expression patterns unaltered. Future experiments should investigate whether this conserved enhancer element mediates reward-related behaviors and cellular adaptations.

In conclusion, we have identified and characterized temporally distinct waves of transcription and chromatin remodeling in striatal neurons. Furthermore, we found that the translation of IEGs is required for activity-dependent chromatin remodeling, particularly at genomic enhancers. Targeted analysis of LRG loci identified a genomic enhancer that is necessary, sufficient, and specific for *Pdyn* transcription. This enhancer is conserved in the human genome and represents a novel therapeutic target for modulation of the kappa opioid receptor system. Continued functional validation of putative enhancer regions within this dataset may provide additional therapeutic targets.

# Materials and methods
## Animals
Male or female adult Sprague-Dawley rats (Charles River, Hartford CT) were co-housed in pairs in plastic filtered cages with nesting enrichment in an Association for Assessment and Accreditation of Laboratory Animal Care-approved animal care facility maintained between 23 and 24°C on a 12-hr light/12-hr dark cycle with ad libitum food (Lab Diet Irradiated rat chow) and water. Bedding and enrichment were changed weekly by animal resources program staff. Animals were randomly assigned to experimental groups. All experiments were approved by the University of Alabama at Birmingham Institutional Animal Care and Use Committee (IACUC) under animal protocol number 20118 or 21306. For primary neuronal cultures, timed pregnant Sprague-Dawley dams were individually housed until embryonic day 18 when striatal cultures were generated as previously described (*Savell et al., 2020*; *Savell et al., 2019a*; *Carullo et al., 2020b*).

## Neuronal cell culture
Cells were maintained in neurobasal media supplemented with B27 and LG for 11–12 days in vitro (DIV) with half media changes on DIV 1, 5–6, and 9–10. For depolarization experiments, neurons were treated with KCl dissolved in neurobasal media to a final concentration of 10 mM for 1 or 4 hr. To test the necessity of protein translation in mediating activity-dependent transcription and chromatin remodeling, anisomycin was dissolved in DMSO and treated to a final concentration of 40 µM for 30 min prior to depolarization. Additional stimuli outlined in *Figure 1—figure supplement 1* included recombinant BDNF (100 ng/ml), the adenylyl cyclase activator FSK (20 µM), and the GABA$_A$ receptor

antagonist gabazine (GBZ, 5 µM). All treatments were applied for 4 hr followed by RNA extraction and RT-qPCR.

## HEK-293T experiments

HEK-293T cells (ATCC CRL-3216; RRID:CVCL_0063) were maintained in Dulbecco's modified Eagle medium + 10% fetal bovine serum and plated at 80,000 cells/well in 24-well plates. 24 hr later cells were transfected with Fugene HD (Promega) and constructs containing dCas9-VPR and gRNA vectors targeting the conserved DAR region (500 ng plasmid DNA in molar ratios sgRNA:dCas9-VPR). Forty-eight hours later, cells were lysed and RNA was extracted.

## CRISPR/dCas9 gRNA design and delivery

Guide RNAs targeting *lacZ*, the rat *Pdyn* enhancer, and conserved human *PDYN* enhancer were designed using CHOPCHOP as previously described (*Savell et al., 2020*; *Duke et al., 2020*; *Savell et al., 2019a*; *Carullo et al., 2020b*; *Carullo et al., 2021*). CRISPR/dCas9 and gRNA constructs were packaged into lentiviral vectors and transduced as previously described (*Savell et al., 2020*; *Duke et al., 2020*; *Carullo et al., 2021*; *Carullo et al., 2020a*; *Savell et al., 2019b*). gRNA sequences can be found in *Supplementary file 4*.

## RNA extraction and RT-qPCR

RNA extractions, cDNA synthesis, and RT-qPCR were performed as previously described (*Savell et al., 2020*; *Savell et al., 2019a*; *Carullo et al., 2020b*; *Savell et al., 2019b*). All RT-qPCR primers can be found in *Supplementary file 4*.

## Western blotting

DIV 11–12 cultured rat embryonic striatal neurons were treated with 40 µM anisomycin followed by 4 hr of depolarization with 10 mM KCl in 12-well plates. Following depolarization, media was removed, and wells were washed with 1× Tris-buffered saline (TBS). Cells were lysed using radio-immunoprecipitation assay (RIPA) lysis buffer (50 mM Tris–HCl pH 8, 150 mM NaCl, 1% NP-40 Substitute, 0.5% sodium deoxycholate, 0.1% sodium dodecyl sulfate [SDS], 1× HALT protease inhibitor cocktail). Following protein separation and transfer, Polyvinylidene difluoride (PVDF) membrane was incubated with FOS primary antibody (Cell Signaling Technology Cat# 2250, RRID:AB_2247211; 1:1000 in Tris-buffered saline with 0.1% Tween 20 (TBST) )and β-tubulin primary antibody (Millipore Cat# 05-661, RRID:AB_309885; 1:2000 in TBST) overnight at 4°C. Following primary antibody incubation, secondary antibodies (LI-COR Biosciences Cat# 926-68071, RRID:AB_10956166; 1:10,000 and LI-COR Biosciences Cat# 926-32212, RRID:AB_62184; 1:10,000 in 1:1 TBST:Intercept blocking buffer with 0.02% SDS) for 1 hr at room temperature. Imaging was performed using a Licor Odyssey imager.

## RNA-seq library preparation and analysis

RNA was extracted from cultured rat embryonic striatal neurons following stimulation (RNeasy, QIAGEN) and submitted to the Genomics core lab at the Heflin Center for Genomic Sciences at the University of Alabama at Birmingham for library preparation as previously described (*Savell et al., 2020*; *Carullo et al., 2020b*). RNA-seq libraries were generated from experiments independent of the ATAC-seq experiments. For the RNA-seq experiment of neurons treated with vehicle or KCl for 1 hr, there were three replicates within the KCl group and four replicates within the vehicle group. For the RNA-seq experiment of neurons treated with vehicle or KCl for 4 hr, there were four replicates within each group (4 Veh, 4 KCl). 75 bp paired-end libraries were sequenced using the NextSeq500. Paired-end FASTQ files were analyzed using a custom bioinformatics pipeline built with Snakemake (*Mölder et al., 2021*) (v6.1.0). Briefly, read quality was assessed using FastQC before and after trimming with Trim_Galore! (*Martin, 2011*) (v0.6.7). Splice-aware alignment to the mRatBn7.2/Rn7 reference assembly for cultured embryonic rat striatal neurons using the associated Ensembl gene transfer format (gtf) file (version 105) and the Hg38 reference assembly using the associated Ensembl gtf (version 99) for previously published human datasets was performed with STAR (*Dobin et al., 2013*) (v2.7.9a). Binary alignment map (BAM) files were indexed with SAMtools (*Li et al., 2009*) (v1.13). Gene-level counts were generated using the featureCounts function within the Rsubread package (*Liao et al., 2019*) (v2.6.1) in R (v4.1.1). QC metrics were collected and reviewed with MultiQC (*Ewels*

*et al., 2016*) (v1.11). Differential expression testing was conducted using DeSeq2 (*Love et al., 2014*) (v1.38.3). DEG testing p-values were adjusted with the Benjamini–Hochberg method (*Benjamini and Hochberg, 1995*). DEGs were designated as those genes with an adjusted p-value <0.05 and a |log2FoldChange| > 0.5. DEGs were calculated by comparing the KCl and Vehicle treatment groups at each respective timepoint. Molecular function and cellular component GO terms were identified by first characterizing upregulated DEGs specific for the 1 or 4 hr timepoint. Ensembl gene IDs for all genes were input into gProfiler (*Raudvere et al., 2019*) with a custom background of all expressed genes (counts >0). Resulting p-values were adjusted with the Benjamini–Hochberg (*Benjamini and Hochberg, 1995*) method.

## ATAC-seq library preparation and analysis

ATAC-seq libraries were prepared as previously described (*Carullo et al., 2020b*). ATAC-seq libraries were generated from experiments independent of the RNA-seq experiments. For the ATAC-seq experiment of neurons treated with vehicle or KCl for 1 hr, there were three replicates within each treatment group (3 Veh, 3 KCl). For the ATAC-seq experiment of neurons treated with vehicle or KCl for 4 hr, there were three replicates within each treatment group (3 Veh, 3 KCl). For the ATAC-seq experiment of neurons pretreated with DMSO or anisomycin, there were 4 replicates within each treatment group (4 DMSO + Veh, 4 DMSO + KCl, 4 anisomycin + KCl). 75 bp paired-end libraries were then sequenced using the NextSeq500 at the Genomics core lab at the Heflin Center for Genomic Sciences at the University of Alabama at Birmingham as previously described (*Carullo et al., 2020b*). Paired-end FASTQ files were analyzed using a custom bioinformatics pipeline built with Snakemake (*Mölder et al., 2021*) (v6.1.0). Read quality was assessed with FastQC before and after trimming (trimming was performed with Trim_Galore! [*Martin, 2011*] v0.6.7). Particularly, Nextera adapters (5'-CTGTCTCTTATA-3') were identified and removed. For rat striatal neuron experiments, FASTQ files were then aligned using Bowtie2 (*Langmead and Salzberg, 2012*) (v2.4.4) to the mRatBn7.2/Rn7 reference assembly Ensembl gene transfer format (gtf) file (version 105). Previously published human datasets were aligned to the Hg38 reference assembly using the associated Ensembl gtf (version 99). Polymerase chain reaction (PCR) duplicates were marked with using Picard (*Broad Institute, 2018*) (v2.26.2). For human ATAC-seq data, encode version 4 of the Hg38 blacklist was used. PCR duplicates, in addition to reads mapping to the mitochondrial genome, were removed using SAMtools (*Li et al., 2009*) (v1.13). BigWig files were generated with deeptools (*Ramírez et al., 2016*) (v3.5.1). QC metrics were collected and reviewed with MultiQC (*Ewels et al., 2016*) (v1.11). For ATAC-seq libraries from cultured rat embryonic striatal neurons, peaks were called using MACS2 (*Zhang et al., 2008*) (v 2.1.1.20160309) callpeak with options - -qvalue 0.01 - -gsize 2626580772 - -format BAMPE. Differential accessibility analysis was performed with Diffbind (*Ross-Innes et al., 2012*) (v3.8.4). DARs were defined as regions with an adjusted p-value <0.05. Reads within peaks were counted using the dba. count() function with options bParallel = TRUE, summits = 250, bUseSummarizeOverlaps = TRUE, and score = DBA_SCORE_TMM_READS_FULL_CPM. Motif enrichment was investigated using HOMER (*Heinz et al., 2010*) (v4.11.1) findMotifsGenome.pl. Motif enrichment was calculated by dividing the total number of motifs identified within a defined bin and then dividing by the bin size (50 bp). This number was further divided by the peak set size (5312). This normalization ensures histograms can be compared even when generated with different bin sizes or peak sets with a significantly higher or lower number of peaks included. Dimensionality reduction of motif enrichment across all DARs was conducted using the umap package in R (v2.10.0) to calculate 10 UMAP (*McInnes et al., 2018*) components with custom options min_dist = 1e-250, n_neighbors = 30, and n_components = 10. UMAP values were used to cluster points with the hdbscan() function of the dbscan (*Hahsler et al., 2019*) package (v1.1–11) with minPts = 150.

## Pseudotime

Pseudotime calculations were performed using Monocle v3_1.3.1 (*Cao et al., 2019*) and Seurat v4.3.0 (*Hao et al., 2021*). A Seurat object containing Drd1-MSNs from male and female adult Sprague-Dawley rats treated with a single cocaine injection (*Phillips et al., 2023*) was subject to the standard dimensionality reduction and clustering workflow using 17 principal components and a resolution value of 0.2. Annotation, counts metadata, and cell barcode information were extracted from the Seurat object and reconstructed into a Monocle v3 cds object using new_cell_data_set(). Rather than

reclustering, the UMAP coordinates were extracted from the Seurat object and added to the Monocle object. The trajectory graph was created using learn_graph() and the root cell was chosen in the 'inactivated' population that expressed *Drd1* and *Htr4* (*Phillips et al., 2023*). Pseudotime values for each cell were exported from the cds object and added back to the Seurat object metadata.

## Gene regulatory network reconstruction

Reconstruction of gene regulatory networks within *Drd1*-MSNs was performed using Epoch (*Su et al., 2022*). Dynamically expressed genes were found using findDynGenes(), and a p-value threshold of 0.05 was used to filter for significance. Transcription factors for *R. norvegicus* were downloaded from AnimalTFDB4.0 (*Shen et al., 2023*) used to construct an initial static network using reconstructGRN(). Epoch uses a Context Likelihood of Relatedness (CLR) model to infer relationships between transcription factors (TFs) and transcription targets (TGs) using transcriptomic information and calculated pseudotime values. A process which Epoch terms as 'crossweighting' was then performed to filter out indirect relationships or non-logical connections. Next, a dynamic network was extracted by fractionating the *Drd1* cells by 'epochs', which is based on pseudotime. TF–TG relationships were ranked using PageRank (*Brin and Page, 1998*).

## Single-nucleus dissociation

Flash-frozen NAc tissue was added to a tube containing 500 µl lysis buffer (1 mM Tris–HCl pH 7.4, 1 mM NAc, 0.3 mM MgCl$_2$, 0.01% Tween-20, 0.01% Igepal in nuclease-free water, 0.001% Digitonin, 0.1% bovine serum albumin [BSA]), homogenized using a motorized homogenizer and RNase-free pestle, and incubated on ice for 5 min. Samples were then pipette mixed 15× and incubated on ice for an additional 10 min. Following lysis, four samples from same sex and treatment were grouped into a single sample, and 2 ml chilled wash buffer (10 mM Tris–HCl pH 7.4, 10 mM NACl, 3 mM MgCl$_2$, 1% BSA, 0.1% Tween-20) was added. Samples were then passed through 70 and 40 µM Scienceware Flowmi Cell Strainers. Samples were then centrifuged at 250 rcf for 5 min at 4°C. Supernatant was removed and the pellet was resuspended in 1 ml of chilled wash buffer (10 mM Tris–HCl pH 7.4, 10 mM NACl, 3 mM MgCl$_2$, 1% BSA, 0.1% Tween-20). 100 µl of sample was then stained with 7-aminoactinomycin D and used to set a representative gate for fluorescence-activated cell sorting (FACS). This gate was then used to purify nuclei further. Following FACS, samples were spun at 250 rcf for 10 min at 4°C to remove any remaining debris. Samples were then loaded into individual wells of the Chromium NextGem Single Cell Chip using four of the eight available wells.

## snATAC-seq alignment and object construction

10× Genomics snATAC-seq libraries were generated from male and female Sprague-Dawley rats exposed injected with 20 mg/kg cocaine (or saline control) once daily for 7 days. snATAC-seq data were aligned to the mRatBN7.2/Rn7 reference assembly using cellranger-atac v2.1.0 and the procedure outlined by 10× Genomics (*Satpathy et al., 2019*). Analysis of the cellranger-atac output was analyzed using Signac v1.9.0 (*Stuart et al., 2021*) and Seurat v4.3.0 (*Hao et al., 2021*). Counts, fragments, and barcode information for each treatment group were loaded and combined into Seurat objects as outlined by the Stuart lab website (https://stuartlab.org/signac/). The groups were then merged into one object based on a common set of features and the standard Signac dimensionality reduction and clustering workflow was performed using 2:30 dimensions and a resolution value of 0.2. Data were then integrated with the snRNA-seq data from the same samples. Common anchors between the two were used to predict and label the cell types present in the snATAC-seq data.

## Co-accessibility analysis

Detection of cis-coaccessible sites in the snATAC-seq data was performed using Cicero v1.3.9 (*Pliner et al., 2018*) and Monocle v3_1.3.1 (*Cao et al., 2019*). The finalized snATAC-seq Seurat object was first converted into a Monocle3 cds object and then converted into a Cicero object. The Cicero connections were found using run_cicero with the Cicero object and a dataframe containing chromosome lengths extracted from the Seurat object. Cis-coaccessible networks were then calculated using generate_ccans(), and the produced links were added back to the starting Seurat object.

## Data availability

All relevant data that support the findings of this study are available by request from the corresponding author (J.J.D.). Custom code can be found at https://github.com/Jeremy-Day-Lab/Phillips_etal_2023,

(copy archived at *Phillips, 2023*). Sequencing data that support the findings of this study are available in Gene Expression Omnibus. Accession numbers of specific datasets are outlined below. Bulk RNA-seq primary rat striatal neurons 1 hr vehicle or KCl: GSE150499. Bulk RNA-seq primary rat striatal neurons 4 hr vehicle or KCl: GSE233752. Bulk ATAC-seq primary rat striatal neurons 1 hr vehicle or KCl: GSE150589. Bulk ATAC-seq primary rat striatal neurons 4 hr ehicle or KCl: GSE233368. Bulk ATAC-seq primary rat striatal neurons 4 hr vehicle or KCl with anisomycin: GSE233368. snATAC-seq adult rat nucleus accumbens: GSE233754.

## Acknowledgements

We thank all current and former Day Lab members for assistance and support. This work was supported by NIH grants DP1DA039650, R01MH114990, R01DA053743, R01DA054714, and the McKnight Foundation Neurobiology of Brain Disorders Award (JJD). LI is supported by the Civitan International Research Center at UAB. RAPIII is supported by the AMC21 scholar program and the UAB T32 in the Neurobiology of Cognition and Cognitive Disorders (T32NS061788). We acknowledge support from the University of Alabama at Birmingham Biological Data Science Core (RRID:SCR_021766), and the UAB Heflin Center for Genomic Sciences.

## Additional information

### Competing interests

Jeremy J Day: Reviewing editor, *eLife*. The other authors declare that no competing interests exist.

### Funding

| Funder | Grant reference number | Author |
|---|---|---|
| National Institute on Drug Abuse | DP1DA039650 | Jeremy J Day |
| National Institute on Drug Abuse | R01DA053743 | Jeremy J Day |
| National Institute on Drug Abuse | R01DA054714 | Jeremy J Day |
| National Institute of Mental Health | R01MH114990 | Jeremy J Day |
| McKnight Foundation | Neurobiology of Brain Disorders Award | Jeremy J Day |
| National Institute of Neurological Disorders and Stroke | T32NS061788 | Robert A Phillips |

The funders had no role in study design, data collection, and interpretation, or the decision to submit the work for publication.

### Author contributions

Robert A Phillips, Conceptualization, Data curation, Formal analysis, Funding acquisition, Investigation, Visualization, Methodology, Writing - original draft, Writing – review and editing; Ethan Wan, Formal analysis, Investigation, Writing – review and editing; Jennifer J Tuscher, Olivia R Drake, Formal analysis, Investigation, Methodology, Writing – review and editing; David Reid, Investigation; Lara Ianov, Data curation, Software, Formal analysis, Visualization; Jeremy J Day, Conceptualization, Data curation, Formal analysis, Supervision, Funding acquisition, Visualization, Writing - original draft, Project administration, Writing – review and editing

### Author ORCIDs

Robert A Phillips (iD) http://orcid.org/0000-0003-3560-4747
Jeremy J Day (iD) http://orcid.org/0000-0002-7361-3399

## Ethics

All experiments were approved by the University of Alabama at Birmingham Institutional Animal Care and Use Committee (IACUC) under animal protocol number 20118 or 21306.

Reviewer #1 (Public Review): https://doi.org/10.7554/eLife.89993.3.sa1
Reviewer #2 (Public Review): https://doi.org/10.7554/eLife.89993.3.sa2
Reviewer #3 (Public Review): https://doi.org/10.7554/eLife.89993.3.sa3
Author Response https://doi.org/10.7554/eLife.89993.3.sa4

# Additional files

## Supplementary files

- Supplementary file 1. Differential expression testing results for 1 and 4 hr timepoints.
- Supplementary file 2. Molecular function gene ontology (GO) terms for 1 and 4 hr specific differentially expressed genes (DEGs).
- Supplementary file 3. Differential accessibility testing results for 4 h timepoint.
- Supplementary file 4. CRISPR sgRNA and RT-qPCR primer sequences.
- MDAR checklist

## Data availability

All relevant data that support the findings of this study have been uploaded as source data files. Custom code can be found at https://github.com/Jeremy-Day-Lab/Phillips_etal_2023, (copy archived at *Phillips, 2023*). Sequencing data that support the findings of this study are available in Gene Expression Omnibus. Accession numbers of specific datasets are outlined here. Bulk RNA-seq primary rat striatal neurons 1 hr vehicle or KCl: GSE150499 Bulk RNA-seq primary rat striatal neurons 4 hr vehicle or KCl: GSE233752 Bulk ATAC-seq primary rat striatal neurons 1 hr vehicle or KCl: GSE150589 Bulk ATAC-seq primary rat striatal neurons 4 hr vehicle or KCl: GSE233368 Bulk ATAC-seq primary rat striatal neurons 4 hr vehicle or KCl with anisomycin: GSE233368 snATAC-seq adult rat nucleus accumbens: GSE233754.

The following datasets were generated:

| Author(s) | Year | Dataset title | Dataset URL | Database and Identifier |
|---|---|---|---|---|
| Carullo NVN, Phillips Iii RA, Simon RC, Soto SAR, Hinds JE, Salisbury AJ, Revanna JS, Bunner KD, Ianov L, Sultan FA, Savell KE, Gersbach CA, Day JJ | 2020 | RNA-seq datasets for enhancer RNA quantification in 'Enhancer RNAs predict enhancer-gene regulatory links and are critical for enhancer function in neuronal systems' | https://www.ncbi.nlm.nih.gov/geo/query/acc.cgi?acc=GSE150499 | NCBI Gene Expression Omnibus, GSE150499 |
| Phillips III RA, Wan E, Tuscher JJ, Reid D, Drake OR, Ianov L, Day JJ | 2023 | RNA-seq datasets for 'Temporally Specific Gene Expression and Chromatin Remodeling Programs Regulate a Conserved Pdyn Enhancer' [RNA-seq] | https://www.ncbi.nlm.nih.gov/geo/query/acc.cgi?acc=GSE233752 | NCBI Gene Expression Omnibus, GSE233752 |
| Carullo NVN, Phillips Iii RA, Simon RC, Soto SAR, Hinds JE, Salisbury AJ, Revanna JS, Bunner KD, Ianov L, Sultan FA, Savell KE, Gersbach CA, Day JJ | 2020 | ATAC-seq datasets for chromatin accessibility quantification in 'Enhancer RNAs predict enhancer-gene regulatory links and are critical for enhancer function in neuronal systems' | https://www.ncbi.nlm.nih.gov/geo/query/acc.cgi?acc=GSE150589 | NCBI Gene Expression Omnibus, GSE150589 |

*Continued on next page*

*Continued*

| Author(s) | Year | Dataset title | Dataset URL | Database and Identifier |
|---|---|---|---|---|
| Phillips III RA, Wan E, Tuscher JJ, Reid D, Drake OR, Ianov L, Day JJ | 2023 | RNA-seq datasets for 'Temporally Specific Gene Expression and Chromatin Remodeling Programs Regulate a Conserved Pdyn Enhancer' | https://www.ncbi.nlm.nih.gov/geo/query/acc.cgi?acc=GSE233368 | NCBI Gene Expression Omnibus, GSE233368 |
| Phillips III RA, Wan E, Tuscher JJ, Reid D, Drake OR, Ianov L, Day JJ | 2023 | Single nucleus ATAC-seq datasets for 'Temporally Specific Gene Expression and Chromatin Remodeling Programs Regulate a Conserved Pdyn Enhancer' | https://www.ncbi.nlm.nih.gov/geo/query/acc.cgi?acc=GSE233754 | NCBI Gene Expression Omnibus, GSE233754 |

The following previously published datasets were used:

| Author(s) | Year | Dataset title | Dataset URL | Database and Identifier |
|---|---|---|---|---|
| Sanchez-Priego C, Hu R, Boshans LL, Lalli M, Janas JA, Williams SE, Dong Z, Yang N | 2022 | Mapping cis-regulatory elements in human excitatory and inhibitory neurons links psychiatric disease heritability and activity-regulated transcriptional programs [RNA-seq] | https://www.ncbi.nlm.nih.gov/geo/query/acc.cgi?acc=GSE196855 | NCBI Gene Expression Omnibus, GSE196855 |
| Sanchez-Priego C, Hu R, Boshans LL, Lalli M, Janas JA, Williams SE, Dong Z, Yang N | 2022 | Mapping cis-regulatory elements in human excitatory and inhibitory neurons links psychiatric disease heritability and activity-regulated transcriptional programs [ATAC-seq] | https://www.ncbi.nlm.nih.gov/geo/query/acc.cgi?acc=GSE196854 | NCBI Gene Expression Omnibus, GSE196854 |
| Yeh SY, Estill M, Lardner CK, Browne CJ, Minier-Toribio A, Futamura R, Beach K, McManus CA, Xu SJ, Zhang S, Heller EA, Shen L, Nestler EJ | 2022 | Cell-type-specific whole-genome landscape of ΔFOSB binding in nucleus accumbens after chronic cocaine exposure | https://www.ncbi.nlm.nih.gov/geo/query/acc.cgi?acc=GSE197668 | NCBI Gene Expression Omnibus, GSE197668 |

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
