## [Editor Report · eLife assessment]

This is an **important** study that uses chromatin accessibility as a measure to determine the impact of neuronal activity on the state of chromatin regulatory elements in striatal neurons. The authors provide **convincing** evidence of how Pdyn gene expression is highly dependent on a distal regulatory genomic region both at basal and upon neuronal activation in this particular system, a mechanism conserved as well in human neuronal cells. Although the basic idea of accessibility changes have been studied before, this paper ties previous findings all together in one place and uses the analysis to identify a functionally relevant and conserved enhancer for the prodynorphin gene with potential relevance for neuropsychiatric disorders beyond basic cellular neuroscience. The study will be of interest to neuroscientists studying the striatum, neuronal plasticity, or related neuropsychiatric disorders.

---

## [Referee Report · Reviewer #1 (Public Review)]

Summary:

In this manuscript the authors use ATAC-seq to find regions of the genome of rat embryonic striatal neurons in culture that show changes in regulatory element accessibility following stimulation by KCl-mediated membrane depolarization. The authors compare 1hr and 4hr transcriptomes to see both rapid and late response genes. When they look at ATAC-seq data they see no changes in accessibility at 1hr but strong changes at 4hr. The differentially accessible sites were enriched for the AP-1 site, suggesting regulation by Fos-Jun family members, and consistent with the requirement for IEG expression, anisomycin blocked the increase in accessibility. To test the functional importance of this regulation the authors focus on a putative enhancer 45kb upstream of the activity-induced gene encoding the neuromodulator dynorphin (Pdyn). To test the function of this region, the authors recruited CRISPRi to the site, which blocked KCL-dependent Pdyn induction, or CRISPRa, which selectively increased Pdyn expression in the absence of KCl. Finally the authors reanalyze other human and rat datasets to show cell-type specific function of this enhancer correlated to Pdyn expression.

The idea that stimuli that induce expression of Fos in neurons can change accessibility of regulatory elements bound from Fos has been shown before, but almost all the data are from hippocampal neurons so it is nice to see the different cell type used here. The most interesting part of the study is the identification of the Pdyn enhancer because of the importance of this gene product in the function of striatal neurons. Overall the conclusions appear to be well supported by the data.

---

## [Referee Report · Reviewer #2 (Public Review)]

This study aims to characterize transcriptional and epigenetic activity-dependent striatal neuronal adaptations using rat primary cultures, a model still poorly characterized up to date. In addition, the authors aim to interrogate regulatory mechanisms that could modulate the expression of a highly-striatal enriched gene responding to neuronal activation in striatal neuronal cells, the Pdyn gene.

Among the major strengths of the article there is the generation of high quality neuronal RNA-seq and ATAC-seq data in rat striatal neuronal cells in basal level and upon neuronal activity, a experimental setup that has not been so characterized as other more common ones such as mouse hippocampal neuronal cells. In this model, the authors clearly demonstrate the need of protein translation to induce the transcriptional waves of late response genes. In addition, the functional characterization of an enhancer of the Pdyn gene might be of great interest for translational applications in which alterations of this gene might be occurring in neurological disorders.

On the other hand, the manuscript presents some limitations to be considered. One of the major points in this regard is that, at least in part, some of the conclusions reached by the study related to the induction of particular transcriptional programs upon neuronal activation, the changes in chromatin state, and the need of protein translation for proper induction of LRGs have been already previously described in the literature, affecting the novelty of the study. However, it is needed to be mentioned, that these previous studies were not conducted using the same model (rat striatal neurons), which can make some differences in the final outputs. The other major cautionary point in the study is the selection of the time point for distinguishing early versus late response genes, as the short difference in time and the overlap of part of the transcriptional signature between them suggest that the transcriptional waves are somehow partially overlapping (also probably in part because of the recurrent stimulation of the primary cultures with KCl), which could result in missing part of the late-response genes.

Despite this, the conclusions raised in the study are well supported by the data generated in it.

In summary, the study presents a useful set of transcriptomic and epigenomic data of activity-dependent striatal neuronal programs in rats, which will be of great use for the scientific community working in this not so well characterized model. In addition, the characterization of a Pdyn distal regulatory genomic region involved in its transcriptional regulation, both at basal levels and upon neuronal activation in this particular system, can present translational relevance for striatal disorders such as Huntington's disease or other neuropsychiatric disorders.

---

## [Referee Report · Reviewer #3 (Public Review)]

This work contributes to the literature characterizing early and late waves of transcription and associated chromatin remodeling following neuronal depolarization, here in cultured embryonic striatum. While they find IEG transcription 1 h after depolarization, they find chromatin remodeling is slower (opening at the 4 h time point). While this is not the first paper to describe chromatin changes in response to neuronal activity, this paper ties previous findings all together in one place using novel sequencing analyses and visualizations. Previous work has found remodeling occurring at the 1 h time point, so the lack of differences at that early time point in the current study needs to be better understood and the "temporal decoupling" described by authors should be further explored. Differences may be due to chromatin at IEG regulatory regions already being open in embryonic tissue (here) vs generally more closed in adult tissue (previous), or due to previous studies using protocols to specifically silence neurons prior to activation. The authors next show that the chromatin remodeling that occurs at the late (4 h) stage is largely in putative regulatory regions of the genome (rather than gene bodies), and is dependent on translation, which validates and extends the prior literature. The authors then transition from genome-wide basic neuroscience to focus on a specific gene of interest, prodynorphin (Pdyn), and a putative enhancer they identify from their chromatin analysis. They target CRISPR-activating and -inhibiting complexes to the putative enhancer and demonstrate that accessibility of this locus is necessary and sufficient for Pdyn transcription. They then show that at least one PDYN enhancer is conserved from rodents to humans, and is only activity-regulated in human GABAergic but not glutamatergic neurons. Finally, the authors generate snATAC-seq and show Pdyn gene and enhancer activity is also cell-type-specific in rat striatum. The Pdyn work, in particular, is thorough and novel, and demonstrates a translational aspect of this work.

---

## [Author Response]

The following is the authors’ response to the original reviews.

We thank all the reviewers for their comments and constructive feedback regarding our manuscript. We have made many changes to strengthen the manuscript, including addition of two new experiments (presented in Fig. S1) that help to clarify the nature and scope of activation of late response genes in striatal neurons. Our specific responses to individual reviewer comments are provided below.

**Reviewer #1**
Public reviewWeaknesses: The timing and the location of the accessibility changes are meaningfully different from other similar studies, which should be discussed. The authors provide good data for the function of a single enhancer near Pdyn, but could contextualize this with respect to other regulatory elements nearby.

In the revised manuscript, we have expanded our discussion of the differences between chromatin accessibility changes observed in this study and those found in prior reports in different systems. These differences are also addressed in extended detail below. Unfortunately, limitations on resources and time prevented a deeper exploration of additional candidate enhancers near the Pdyn locus. However, we believe our efforts to characterize an activity-dependent enhancer in the Pdyn locus provides a useful starting point, and future studies may seek to more completely define the contributions of nearby regulatory elements.

Recommendations For The Authors1. At 1hr after stimulation in previous papers (Su 2017 which is reference #8 of FernandezAlbert Nat Neurosci. 2019 October ; 22(10): 1718-1730.) there are large increases in accessibility directly over the IEGs, consistent with the concerted transcription of these genes following stimulation. It is surprising that the authors do not see this here, either at 1hr or at 4hr. This difference in results needs to be addressed.

We thank the reviewer for bringing this discrepancy to our attention. Indeed, Su et al. 2017 and Fernandez-Albert et al. 2019 both describe increases in chromatin accessibility at IEG promoters. There are several experimental differences that could be contributing to differences between our study and previously published studies. Two major reasons include the developmental timepoint of the tissue/cells and the cell type/brain region that is being assayed. Su et al. assayed chromatin accessibility in ex vivo slices containing the dentate gyrus from adult mice, while Fernandez-Albert et al. assayed chromatin accessibility in forebrain principal neurons of adult mice following kainic acid injection. Bulk ATAC-Seq experiments described in the present manuscript were generated from cultured embryonic rat striatal neurons. Additionally, baseline chromatin accessibility seems to be significantly different between forebrain principal neurons studied in Fernandez-Albert et al. 2019 and the current study. For example, in Figure 3a of Fernandez-Albert et al. 2019, the Npas4 gene body is not accessible in a saline treated animal. In vehicle treated, cultured embryonic rat striatal neurons, the Fos gene body and associated enhancers are accessible at baseline (Fig. S3), and do not increase with KCl depolarization.

We have expanded our discussion of this discrepancy in the discussion section of the revised manuscript, and included additional citations addressing this difference.

1. It is also somewhat surprising that the authors see almost no regions that show changes in accessibility at 1hr and then a very large number of differentially accessible regions at 4hr. This is quite different from the more rapid changes shown for example in Figure 7f in the human GABA neurons even though these are also studies in culture with rapid calcium channel opening. Can the authors speculate on the reason for the difference?

Many previously published studies that use cultured neurons include a pre-treatment in which spontaneous neuronal activity is inhibited with the sodium channel blocker tetrodotoxin (SanchezPriego et al. Cell Reports, 2022; Kim et al. Nature, 2010; Malik et al. Nature Neuroscience, 2014). The Sanchez-Priego et al. Cell Reports manuscript also blocked NMDA receptor activity with the competitive NMDAR antagonist D-AP5 for 12 hours prior to depolarization. Rapid changes in chromatin accessibility observed in other studies at <1 hour timepoints could be due to prior silencing of the cells and subsequent reduction in the accessibility and transcriptional activity of IEGs. Decreased baseline accessibility and transcriptional activity of IEGs can be observed in Figure 1a of Malik et al. 2014, which displays ChIP-Seq tracks for both RNA pol II and H3K27ac. At baseline, H3K27ac and RNA pol II enrichment is low throughout the Fos locus. Subsequent depolarization of silenced neurons drives accessibility and transcription of the Fos gene and associated enhancers. In contrast, we found accessible chromatin at Fos enhancer elements at baseline (without stimulation; Fig. S3).

The experiments described in the current study do not include any pre-treatment with tetrodotoxin or D-AP5, and thus the neurons are expected to be spontaneously active. This baseline electrophysiological activity may result in increased accessibility and transcription at IEG loci, which ultimately makes it difficult to identify activity-dependent increases in IEG accessibility at timepoints <1 hour. Furthermore, a previously published manuscript from our lab (Carullo et al. Nucleic Acids Research, 2020) conducted ATAC-seq on cultured embryonic rat cortical, hippocampal, and striatal neurons and found that transcribed enhancers for IEG loci (including Fos) had decreased chromatin accessibility following depolarization when compared to vehicle treatment. These differences in experimental design (including cell type, model organism, developmental timepoint, and treatment paradigm) may all contribute to differences in the temporal dynamics of chromatin remodeling between the current manuscript and previously published studies.

1. Experimentally it can be challenging to repress a single enhancer and show a significant effect on gene regulation which makes the repression in Fig 6c somewhat unexpected. There are several regions near Pdyn that show activity-dependent changes in accessibility in the human cells (Fig. 7e) and presumably in the rat neurons too (Fig. 5a shows a few but most of the intervening region is cut out). Did the authors target any of these other regions?

We chose the identified regulatory element upstream of the Pdyn TSS because it met several criteria that we determined are important for characterizing LRG enhancers. These criteria are outlined in the Results: “(1) located in non-coding regions of the genome, (2) inaccessible at baseline and accessible following depolarization, and (3) inaccessible when depolarization was paired with protein synthesis inhibition.” Indeed, ATAC-seq experiments presented in the current study demonstrate that thousands of genomic regions undergo reprogramming, and many of these regions meet these criteria (including additional loci near Pdyn). However, we lacked the time and resources to systematically investigate all other enhancers, and did not target any other regions within the Pdyn locus. While many enhancers may regulate a single gene, the identified enhancer seems to be particularly important for activity-dependent Pdyn gene expression. Importantly, CRISPRi-based repression of this enhancer (Fig. 6c) did not reduce basal Pdyn expression as compared to a non-targeting control, but completely blocked stimulus-dependent induction of Pdyn transcription. We believe this is a useful starting point, and future studies may seek to more completely define the contributions of nearby regulatory elements.

1. The authors should clarify in the methods or figure legends the number of independent replicate libraries for each experiment and were the RNA and ATAC libraries made from the same or different experiments.

We thank the reviewer for bringing this to our attention. We have clarified the number of replicates in the methods as outlined below. Additionally, RNA and ATAC libraries were generated from different experiments, and this information is also now included in the methods.

Within the ATAC-Seq library preparation and analysis methods section: “ATAC-seq libraries were generated from experiments independent of the RNA-seq experiments. For the ATAC-seq experiment of neurons treated with vehicle or KCl for 1 h, there were 3 replicates within each treatment group (3 Veh, 3 KCl). For the ATAC-seq experiment of neurons treated with vehicle or KCl for 4 h, there were 3 replicates within each treatment group (3 Veh, 3 KCl). For the ATAC-seq experiment of neurons pre-treated with DMSO or Anisomycin, there were 4 replicates within each treatment group (4 DMSO + Veh, 4 DMSO + KCl, 4 Anisomycin + KCl).”

Within the RNA-seq library preparation and analysis methods section: “RNA-seq libraries were generated from experiments independent of the ATAC-seq experiments. For the RNA-seq experiment of neurons treated with vehicle or KCl for 1 h, there were 3 replicates within the KCl group and 4 replicates within the vehicle group. For the RNA-seq experiment of neurons treated with vehicle or KCl for 4 h, there were 4 replicates within each group (4 Veh, 4 KCl).”

**Reviewer #2**
Public reviewFirst of all, at a conceptual level, most of the findings related to the induction of particular transcriptional programs upon neuronal activation the changes in chromatin state, and the need for protein translation for proper induction of LRGs have been broadly characterized previously in the literature (Tyssowski et al., Neuron, 2018; Ibarra et al., Mol. Syst. Biol., 2022; and also reviewed by Yap and Greenberg, Neuron, 2018). In addition, it is not so obvious why to focus on Pdyn gene regulatory regions among the thousands of genes upregulated and with modified chromatin landscape after neuronal activation. The authors highlight three particular traits of this gene as the reason to choose it, but those traits are probably shared by most of the genes that are part of the LRGs set.

We thank the reviewer for these comments, and have included these important publications as citations in our manuscript. With over 5,000 differentially accessible chromatin regions following KCl stimulation, it was not possible to follow up on all regulatory regions or linked genes in a rigorous way. Therefore, we selected a target candidate enhancer near the Pdyn locus for mechanistic studies. In addition to the criteria highlighted in the manuscript, we chose this locus due to decades of literature establishing the importance of prodynorphin in the striatum, and the role of this gene in human neuropsychiatric diseases. We would argue that this increases the relevance of more detailed exploration of this gene, and makes our results applicable to a broader pre-existing literature.

At the methodological level, some attention should be put into the timings chosen for generating the data. The authors claim that these time points (1h and 4hrs) identify the first (i.e IEGs) and second (i.e LRGs) waves of transcription. However, at 4hrs the highest over-expressed genes are still IEGs, as shown in the volcano plots of Figure 1B and 1C, showing a high overlap with up-regulated genes found at 1h (Figure 1D). This might suggest that the 4hrs time point is somewhere in between the first and second wave of transcription, probably missing some of the still-to-be-induced LRGs of the latest one.

Given that the depolarization applied in RNA-seq and ATAC-seq experiments is continuous, it was not unexpected to find IEGs present at both 1 h and 4 h timepoints. The revised manuscript contains a new experiment (Fig. S1d-f) demonstrating that a shorter depolarization period (1 h KCl followed by a 3 h wash off period) also induces Fos mRNA, but to a much lower extent than 4 h continuous stimulation. In contrast, both short (1 h) and long (4 h) depolarization periods induce Pdyn to equivalent levels when measured at 4 h after the onset of the stimulus. These experiments support our conclusion that LRGs require a temporal delay, and not simply extended stimulation. Nevertheless, the reviewer is correct that a 4 h timepoint may potentially miss some LRGs that are induced even later. We plan to explore the full timecourse of LRG induction in future studies.

Finally, while only prosed as a suggestion, the assumption that from the data generated in this article, we can envision a mechanism by which AP-1 family of transcription factors interacts with the SWI/SNF chromatin remodeling complex is going too far, as no evidence is provided implicated SWI/SNF in the data presented in the manuscript.

While speculative in the current context, we felt that it was important to highlight this prior literature to identify potential mechanisms that may link IEGs (specifically, AP-1 members) to chromatin remodeling machinery. We have altered this section of the discussion to emphasize that this link is speculative in the context of neuronal chromatin remodeling.

**Recommendations For The Authors**
1. I couldn't find the number of replicates generated for each dataset, neither for RNA nor for ATAC-seq. It could be worth adding these data to the figure legends or in the material and methods.

We thank the reviewer for bringing this to our attention. The number of replicates generated for each dataset are now included in the methods section (see response to Reviewer #1, comment #4 above).

1. In Figure 1D, Gene Ontology terms appear significant only for each of the individual datasets. While this might be expected for the 1h time-point, the 4hrs time-point comprises a big extent of the genes up-regulated at 1h as well, and it is surprising no term related to chromatin or transcription regulation appears as significant. Is this due to the fact that the analysis has been conducted with two separated lists of genes and only the top terms are shown without crossing the data? This could be misleading for the reader and maybe a comparative GO term analysis might be better such as using CluterProfiler or similar tools, that might allow for real comparison of terms enriched in each dataset.

We thank the reviewer for pointing this out. For Figure 1d, GO term analysis was conducted with two separated gene lists, each consisting of timepoint-specific upregulated DEGs. Thus, 772 genes were included for the analysis of 4 h GO terms and 39 genes were included for the analysis of 1 h GO terms. Previously, comparisons of cellular component GO terms included in the current study only included the top 10 GO terms. The revised manuscript contains an updated analysis that compares all enriched GO terms and identifies that three of the top 10 cellular component GO terms for the 1 h gene set are also identified as significantly enriched in the 4 h gene set. We have revised the graph in Fig. 1f to reflect this updated analysis. Overall, our conclusions (that 1 h and 4 h DEG sets fall into distinct functional categories) remains supported by this analysis.

1. In Figure 3D, the graphs show the density of motifs within the DARs in units of"Motifs/Kb/peak" while the x-axis represents the peaks coordinates from -500bp to +500bp. It is not clear to me how this graph is generated and how within 1000bp the profiles can reach values of 18-20 Motifs/Kb/peak. Could this be clarified?

The motif enrichment score was calculated by identifying the number of total motifs within defined 50bp genomic bins surrounding the center of the DAR regions. HOMER builds enrichment histograms that normalize motif presence to set size (e.g., number of peaks or DARs), and also to genomic space (base pairs). While HOMER’s default histogram represents motifs/bp/peak, we converted this to motifs/kb/peak for ease of interpretation. However, to avoid confusion we have returned the y axis labels to the default HOMER settings (motifs/bp/peak). The normalization and units for this graph have been clarified in the methods section.

1. In Figure 4C the newly generated ATAC-seq data is just "targeted" analyzed, showing global tendencies are maintained between the initial generated data and this one. It could be interesting, however, to see the number of DARs obtained in these conditions, especially to see if some DARs are observed in the Anisomycin condition that might be translation-independent.

The experiment described in Figure 4 was designed to both validate the 5,312 DARs and understand the role of protein translation in activity-dependent chromatin remodeling. One way to begin identifying translation-independent DARs is to compare the DMSO + Vehicle group to the Anisomycin + KCl group. With this comparison, any 4 h DAR that has increased accessibility in the Anisomycin + KCl group may be translation-independent as pretreatment with anisomycin did not prevent chromatin remodeling. After conducting this analysis, we identified a very small percentage (3.44%) of 5,312 4 h DARs that still exhibited significantly increased accessibility when pre-treated with Anisomycin. This small number is consistent with the robust effects of anisomycin on KCl-dependent remodeling shown in Fig. 4c-d. However, to confirm that these were in fact translation-independent activity-regulated DARs, we would need to perform direct comparison of chromatin accessibility between neurons pre-treated with Anisomycin and then treated with either vehicle or KCl. Since we did not include an anisomycin only group in experiments in Fig. 4, we cannot confidently claim whether this 3.4% of DARs are translationindependent. Nevertheless, we agree with the reviewer that this is an interesting avenue of future exploration.

**Reviewer #3**
Public review1. Throughout the paper, the authors emphasize a "temporal decoupling" of transcriptional and chromatin response to depolarization, based on a lack of significant chromatin changes at 1h, despite IEG transcription. However, previous publications show significant chromatin remodeling at 1h (e.g. Su et al., NN 2017 in adult dentate gyrus) or 2h (Kim et al., Nature 2010; Malik et al., NN 2014 in cultured embryonic cortical neurons). The discussion briefly mentions this contrast, but it remains difficult to conclude decisively whether there is temporal decoupling when such decoupling is not found consistently. If one is to make broad conclusions about basic neural chromatin response to depolarization, it would be ideal to know under which conditions there is temporal decoupling, or if this is a region-specific phenomenon.

Indeed, prior studies referred to in our manuscript have identified chromatin remodeling at earlier timepoints than we identified here. As addressed above (Reviewer #1, comments 1 & 2), it is possible that this discrepancy arises due to the difference in experimental model system, differences in the type of stimulation applied, pretreatment protocols used to silence neurons prior to activation, or even differences in developmental stage. Differences in each of these parameters make it difficult to make straightforward comparisons between datasets and results in this manuscript. It is possible that other cell types induce IEGs more quickly (or more robustly) in response to stimulation, which could lead to earlier chromatin remodeling. However, the common patterns of chromatin reorganization (e.g., the fact that changes are enriched at AP-1 motifs and are found in intergenic regions at putative enhancers) lend support for the idea that the transcriptional waves identified here can also be found in other cell types and in other contexts.

1. The UMAP analysis is a novel way to probe transcription factor enrichment, but it's unclear what this is actually showing. The authors sought to ask whether "DARs could be separated based on transcription factor motifs in these regions." However, the motifs present in any genomic stretch are fixed based on genomic sequence, so it seems like this analysis might be asking whether certain motifs are more likely to be physically clustered together in the genome, in activity-regulated regions (rather than certain transcription factors acting in concert, as is implied in discussion). While still potentially interesting, this analysis does not seem to give much additional insight into activity-dependent chromatin remodeling beyond the motif enrichment analysis already performed. Nevertheless, to draw stronger conclusions, it would be necessary to compare clustering to a random set of genomic regions of the same length/size to interpret the clustering here. It would also be useful to know whether the ISL1 motif is also enriched in ubiquitously accessible genomic regions in the striatum (and not just DARs).

We agree that additional analysis is needed to explore enrichment of various transcription factor motifs and activity at differently accessible regions of the genome. The motif enrichment analysis in Figure 3 demonstrated the types of motifs that were enriched in DARs (Fig. 3a-c), the overall degree of enrichment (Fig. 3c), and the distribution of those motifs across DAR sites (Fig. 3d). This analysis allowed us to understand whether motifs for cell-defining transcription factors like ISL1 are enriched uniquely in DARs, or are also found in other regions that are accessible at baseline (see direct comparisons between vehicle/baseline peaks and DARs in Fig. 3d). However, these approaches represent enrichment across all DARs as group, and do not show TF presence/absence at any specific DAR. The UMAP analysis presented in Figure 3e allowed identification of DAR clusters based on the presence or absence of specific transcription factor motifs, and allowed us to represent specific DARs in a reduced two-dimensional space. Because this analysis identifies the existence of distinct motifs within single DARs, it allowed us to speculate as to the possibility of transcription factor cooperation within DARs, or the meaning of DAR clusters that appear to be defined by specific motifs (e.g., KLF10 in Fig. 3f). Given the information that this adds to the initial analyses, we argue that its inclusion in the manuscript is useful and potentially informative for generating follow-up hypotheses.

1. The authors identify late-response gene enhancers by 3 criteria. However, only Pdyn was highlighted thereafter. How many putative DARs met these three criteria in striatum? Only Pdyn?

As illustrated in Figures 2 and 4, nearly all of the DARs in our dataset met these criteria, which included presence in non-coding genomic regions, increase in accessibility following stimulation, and prevention of chromatin accessibility changes by protein synthesis inhibition. We did not mean to indicate that the Pdyn locus was unique in this way. In addition to the criteria highlighted in the manuscript, we chose this locus due to decades of literature establishing the importance of prodynorphin in the striatum, and the role of this gene in human neuropsychiatric diseases. We would argue that this increases the relevance of more detailed exploration of the regulator mechanisms that control expression of this gene, and makes our results applicable to a broader pre-existing literature. The revised manuscript includes additional experiments that examine Pdyn expression changes in response to different stimuli, which help to justify the focus on this gene from the beginning of the manuscript.

Recommendations For The Authors1. Figure 1 volcano plots show a scatter primarily in the up-regulated portion at both the 1-h and 4-h time points. However, the Venn diagrams show largely similar numbers of up- and downregulated genes at the 4-h time point. Is the clustering of down-regulated genes tighter/more overlapping? If so, semi-translucent volcano dots or some acknowledgment of the visual discrepancy would be useful.

We thank the reviewer for bringing this to our attention. Down-regulated genes are clustering tighter on the volcano plot due to smaller fold changes. This visual discrepancy is acknowledged by the numeric indicators of up- and down-regulated genes in the upper left-hand corner of the volcano plot.

1. Methods for RNA and ATAC seq analysis align to human genome Hg38, rather than rat?

RNA- and ATAC-Seq analyses from rat neurons were aligned to the mRatBn7.2/Rn7 rat genome. RNA- and ATAC-Seq analyses from human neurons were aligned to the Hg38 human genome. We have updated the methods to make this clear.

1. The introduction states that different classes of neurons induce distinct LRGs. Please add a citation. Citations are also needed for the last statement WRT consequences of chromatin remodeling near LRGs not being concretely linked to LRG transcription.

We thank the reviewer for pointing this out. The revised manuscript now includes additional citations supporting each of these statements.

1. Specify somewhere in Methods that DEGs were compared to vehicle for both 1-h and 4-h (and not 4 vs 1 h).

We thank the reviewer for bringing this to our attention. We have updated the methods to include: “DEGs were calculated by comparing the KCl and Vehicle treatment groups at each respective timepoint.”

1. In Figure 2E, why are the enrichments exactly opposite, especially given these are two different types of input (all baseline peaks vs DARs)?

Odds ratios were calculated by comparing baseline peaks (i.e., ATAC-seq peaks identified in vehicle treated cells) to KCl-induced DARs. This allowed us to identify the enrichment of DARs in specific genomic annotations in comparison to the genomic features that are accessible at baseline, rather than making comparisons to random probe sets or genomic space dedicated to these distinct annotations. This analysis identified that relative to baseline peaks, DARs are significantly depleted in coding regions of the genome and enriched in non-coding regions of the genome. However, given this analysis we agree that it does not make sense to graph both the vehicle (baseline) and DARs on this graph, given that enrichment of each set is determined relative to the other (creating the reciprocal enrichment in this panel). We have updated Fig. 2e to only include points for 4 h DARs.

1. Some references are off. One that I noted was "...chromatin remodeling in the mouse dentate gyrus following 1 h of electricoconvulsive stimulation" should be Su et al 2017 not Malik 2014. For the statement that IEGs are critical regulators of non-neuronal IEGs, the authors may want to add Hrvatin 2017 ref.

We thank the reviewer for bringing this to our attention. We have revised the manuscript to include the correct citation for this claim, and also to incude the Hrvatin, et al reference.

1. It would be helpful for the authors to write out the whole gene name for Pdyn somewhere.

We have updated the text to include the gene name for Pdyn, both in the abstract and also in the introduction of the manuscript.

1. Figure 5f: For ease, please include what is high vs low in the figure caption in addition to the main text.

We thank the reviewer for bringing this to our attention. We have updated the figure caption and main text to include what is high vs low in Pseudotime estimates in Fig. 5f.

1. How are the tracks ordered in Fig8c?

Tracks within Fig. 8c demonstrate snATAC-seq signal at the Pdyn gene locus for transcriptionally distinct cell types within the NAc. The tracks are ordered by cluster size (nuclei number) in the snATAC-seq dataset.